# CAN TEXT-TO-VIDEO MODELS GENERATE REALISTIC HUMAN MOTION?

MOVO: A BENCHMARK FOR EVALUATING HUMAN MOTION REALISM IN TEXT-TO-VIDEO GENERATION

## ABSTRACT

Recent advances in text-to-video (T2V) generation have yielded impressive progress in resolution, duration, and prompt fidelity, with models such as Pika, Gen-3, and Sora producing clips that appear compelling at first glance. Yet, in everyday use and public demos, generated people often "look right but move wrong," exhibiting artifacts like foot sliding, joint hyperextension, and desynchronized limbs. Such failures are not cosmetic: 1) unsafe motions can be copied by viewers, especially juveniles, raising injury risks; 2) in clinical and sports contexts, implausible kinematics corrupt analytics for angle, cadence, and phase, causing misdiagnosis and unsafe return-to-play; and 3) in simulation pipelines, non-physical motion distributions contaminate training and evaluation, degrading sim-to-real transfer. However, existing benchmarks remain inadequate: 1) they lack kinematics awareness, rewarding visual resemblance while joint trajectories violate physiological ranges; 2) they lack rhythm- and body-level temporal metrics, overlooking gait-cycle timing, symmetry, and inter-limb coordination; and 3) they fail to disentangle camera from body motion, letting pans and zooms mask biomechanical errors. To address these gaps, we present **Movo**, the first kinematics-centric benchmark for T2V motion realism. Movo unifies three components: 1) a posture-focused dataset with camera-aware prompts that isolate representative upper- and lower-body actions; 2) skeletal-space metrics, Joint Angle Change (JAC), Dynamic Time Warping (DTW), and Motion Consistency Metric (MCM), that operationalize biomechanical plausibility across joints, rhythms, and constraints; and 3) human validation studies that calibrate thresholds and show strong correlation between skeletal scores and perceived realism. Evaluating 14 leading T2V models reveals persistent gaps: some excel in specific motions but struggle with cross-action consistency, and performance varies widely between open-source and proprietary systems. Movo provides a rigorous, interpretable foundation for improving human motion generation and for integrating biomechanical realism checks into model development, selection, and release workflows. The code and scripts are available at Supplementary Material.

## 1 INTRODUCTION

Text-to-video (T2V) systems have made striking gains in resolution, duration, and prompt following (Wu et al., 2023; Blattmann et al., 2023b; Ho et al., 2022a; Singer et al., 2022; Luo et al., 2023; Wang et al., 2023b; Xing et al., 2023a; Wang et al., 2023a; Esser et al., 2023; An et al., 2023; Chen et al., 2023b; Zhang et al., 2023b; Xing et al., 2023b; Fei et al., 2023; Ho et al., 2022b; Gu et al., 2023; Wang et al., 2023f;c; Zhang et al., 2023a; Zhao et al., 2023; Qiu et al., 2023; Li et al., 2023; Ge et al., 2023; Chen et al., 2023a;c). Models such as Pika, Gen-3, and Sora (Pika, 2024; Runway Research, 2024; OpenAI, 2024) often produce clips that look compelling at first glance. Over the past year, text-to-video has moved from niche demos to mass distribution. Runway raised $308M$ at $3B valuation, while YouTube integrated Google's Veo 3 (Sharma et al., 2025) directly into Shorts, placing prompt-to-video generation inside a product that now averages 200 billion daily views, which is such a step change in reach for synthetic video. Applications of T2V systems are already visible. Many creators monetize generative videos on platforms like TikTok and YouTube Shorts (Hu, 2024; Zhang, 2023), turning synthetic clips into ad revenue at scale. Meanwhile, researchers employ

Figure 1: Overview of the Movo benchmark for evaluating human motion realism in text-to-video generation. The benchmark assesses lower- and upper-body movements (e.g., deadlift, side leg raise, hand punch, waist twist). Videos are collected or recorded, labeled, and used to create prompts. Outputs from open-source and proprietary models are evaluated with Joint Angle Change (JAC), Dynamic Time Warping (DTW), and Motion Consistency Metric (MCM). Human validation includes data preparation, pairwise comparison, and annotation.

generated videos in simulation experiments, from robotics training to controlled behavioral studies, where synthetic footage offers safe and reproducible environments (Qin et al., 2024).

In everyday use and public T2V demos, people frequently "look right but move wrong." Typical artifacts include foot sliding during supposed stance, joint hyperextension, discontinuous velocities, desynchronized upper–lower limbs, and props or body parts that break contact constraints (Louis et al., 2025). These are not cosmetic glitches, they carry real consequences. 1) In the short video settings, viewers may copy faulty motions which raise injury risk, especially for juveniles who are pervasively exposed to online videos but lack the motor control and judgment to detect unsafe form (Kianifar et al., 2017). 2) In clinical pre-screening, rehab, and sports assessment, implausible motion corrupts analytics for angle, cadence and phase. causing misclassification, poor prescriptions, delayed gait-issue detection, and unsafe return-to-play (e.g., masked fall risk), with downstream reinjury, unnecessary imaging, and liability (Nakano et al., 2020; Louis et al., 2025). 3) In simulation and synthetic-data pipelines either in industries or labs, non-physical motion distributions contaminate training and evaluation, worsening sim-to-real transfer and negatively affecting industrial production as well as academic research (Doersch & Zisserman, 2019). 4) For platforms and policy, unrealistic human motion complicates quality gates and disclosure, leading to under-disclosure, unjustified fines and takedowns, viral misuses, likeness-rights disputes, and trust erosion (YouTube, 2024; TikTok, 2024; European Union, 2024). Therefore, the takeaway is simple: "looking like" the action is not enough. We must measure whether generated people move in a biomechanically plausible way and integrate such checks into model selection and release workflows.

General-purpose leaderboards emphasize breadth, overall aesthetics, text–video alignment, optical-flow smoothness, and sometimes action recognition, but they miss three things that matter for human motion. 1) First, lack of kinematics awareness. Pixel or semantics metrics commonly used in T2V benchmarks reward clips that resemble "walking" while joint trajectories violate physiological ranges, exhibit abnormal angle amplitudes, or break inter-limb phase relationships. In some specific domains, decisions are made on joints, angles, and phases. When those are implausible, smooth-looking videos still produce wrong conclusions (Huang et al., 2024; Liu et al., 2023). 2) Second, lack of rhythm-aware and body-level temporal metrics. Common smoothness proxies such as optical flow consistency and warping error quantify frame-to-frame pixel continuity but not gait-cycle timing, symmetry or cadence. Without rhythm-sensitive measures, periodic behaviors can drift in tempo or exhibit off-phase coordination yet still score well on flow-based metrics (Liao et al., 2024; Alfarano et al., 2024). 3) Third, lack of camera-motion disentanglement. Many existing T2V benchmarks operate in raw pixel space, so pans, zooms, and shake confound temporal signals

and can mask contact errors, rigid-body violations, bone-length instability, and abnormal velocities or accelerations. Without body-centric stabilization or skeletal-space analysis, metrics are contaminated by camera motion rather than body dynamics. Methods that "pass" such tests often yield unstable pose estimates and unreliable downstream analytics (Kocabas et al., 2024; Ye et al., 2023).

To address these, we introduce Movo, a kinematics-centric benchmark that asks whether generated people move plausibly, not just look plausible. Movo directly addresses the three gaps above. 1) Posture-focused dataset with camera-aware prompts. To reduce confounds and isolate human motion, we cover representative lower-body and upper-body actions with prompt templates that discourage gratuitous camera motion and keep the mover in focus. 2) Skeletal metrics that operationalize biomechanical realism: JAC (Joint Angle Change) quantifies joint-angle trajectories relative to typical ranges and checks plausible evolution over time—making the evaluation kinematics-aware. DTW (Dynamic Time Warping) on pose dynamics measures temporal phasing and rhythm alignment—capturing cadence and inter-limb timing beyond pixel smoothness. MCM (Motion Consistency Metric) enforces constraint-aware consistency, foot–ground contact, velocity/acceleration continuity, and bone-length stability, so camera motion cannot hide structural violations. 3) Human validation that calibrates thresholds. We conduct pairwise preference studies showing Movo's skeletal scores correlate with perceived motion realism, enabling actionable quality gates that align with emerging platform policies for realistic synthetic depictions. Using Movo, we extensively evaluate 14 leading T2V models, including 8 open-source and 6 propriety solutions. Our findings reveal that while some models excel in specific tasks, such as hand rotations, they struggle to maintain consistent quality across diverse motion types. Performance scores vary significantly, highlighting the need for specialized strategies to improve human motion generation.

## 2 RELATED WORK

### 2.1 TEXT-TO-VIDEO GENERATION DATASET

Text-to-video generation has advanced significantly, supported by various datasets. MSR-VTT dataset(Xu et al., 2016) provides 10,000 videos paired with textual annotations, allowing open-domain video description but not focusing on human motion. InternVid dataset (Wang et al., 2023d) scales multimodal data with more than 7 million videos but focuses on general scenarios rather than specific human actions. Recent works like the EvalCrafter dataset (Liu et al., 2024a) and the VideoFactory dataset (Wang et al., 2023a) aim to improve the quality and alignment of text-to-video generation but still lack data sets centered on human motion. The existing UCF101 dataset (Soomro, 2012) focuses on human action recognition with 101 action classes but lacks textual descriptions, which limits its use for generative tasks. In contrast, our proposed Movo dataset is the first text-to-video generation dataset to focus on human motion. It offers detailed textual descriptions of dynamic movements, filling a crucial gap in generating motion-driven videos, and enabling advances in applications like virtual reality and animation.

### 2.2 TEXT-TO-VIDEO GENERATION MODEL

In recent years, text-to-video generation has made remarkable progress, driven by advances in generative models and the increasing availability of computational resources. The early text-to-vision methods relied primarily on Generative Adversarial Networks (GANs) (Balaji et al., 2019; Skorokhodov et al., 2022; Tulyakov et al., 2018; Wang et al., 2020; 2023e) and Variational Autoencoders (VAEs) (Van Den Oord et al., 2017), demonstrating the feasibility of video generation within simple closed set domains (Gupta et al., 2018; Li et al., 2018; Liu et al., 2019). However, these methods struggle to generate videos in more complex contexts (Wang et al., 2023a). The latest breakthroughs in generative AI has progressed from tokenized Transformer pipelines (Hong et al., 2022; Villegas et al., 2022; Wu et al., 2021; 2022) to diffusion-based models that deliver higher fidelity under practical compute (Ho et al., 2022b; Blattmann et al., 2023b; Singer et al., 2022). Controllability has improved via structural conditioning and planning (Wang et al., 2024b; Lin et al., 2023; Wu et al., 2023). Scaling with Diffusion Transformers further advances quality (Peebles & Xie, 2023; Bao et al., 2023; Gao et al., 2023), inspiring systems such as Latte and Sora (Ma et al., 2024; OpenAI, 2024). See Appendix E for an extended survey.

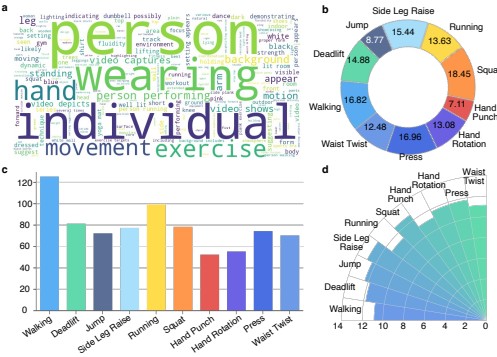

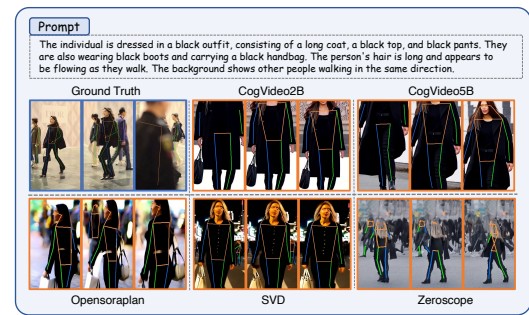

(a) Statistics of video and prompt data: 1) word cloud; 2) avg. duration per movement; 3) sentence count per category; 4) avg. words per sentence.

(b) Comparison of generation results from different models on the "black outfit walking" prompt.

Figure 2: From data to outputs: corpus statistics and model generations on a walking prompt

## 3 POSTURE DATASET

The aim of our posture dataset is to introduce a new and challenging benchmark for the action understanding community. In previous research, most existing fitness datasets (Fieraru et al., 2021; Verma et al., 2020; Zhao et al., 2022) amalgamate various activities without clear distinctions. A primary challenge in constructing the posture dataset lies in developing a systematic taxonomy to organize diverse human activities. We present a more detailed categorical lexicon that includes various possible body postures below the neck.

### 3.1 TAXONOMY

**Classification.** For the first level, we adopt the approach suggested by Humman (Cai et al., 2022), which categorizes activities based on the primary muscles involved. However, given the large number of fine-grained muscles in the human body and the fact that a single activity can engage multiple muscle groups, we consulted with kinesiologists to streamline these categories. As a result, we decided to simplify the activity categories into two main groups: upper body activities (e.g., pressing, hand rotation) and lower body activities (e.g., squatting, jumping), to provide a clearer classification of different types of activity and better align with the synergistic functions of muscle groups in real activities, as shown in Table 3. Although most physical activities engage multiple body regions (e.g., deadlifting involves both the lower and upper body), our classification is based solely on the primary regions responsible for the movement. This focus is particularly relevant for our benchmark, which evaluates whether the movements are executed correctly. For instance, some video generation models produce outputs where, from the camera's perspective, only the leg movements are shown during running. By categorizing activities according to their main active body regions, our taxonomy provides clearer guidance for evaluation.

**Physical Activity.** Building on the primary body regions from the first level, the second level categorizes activities into ten specific exercise groups, encompassing the 10 common physical activities shown in Table 3. These activities were selected because they represent typical movement patterns found in both daily life and fitness settings, and they clearly demonstrate the distinct movement mechanics of the upper and lower limbs. For instance, the *Side Leg Raise* activity primarily engages lower body muscle groups, including the gluteus maximus, gluteus medius, and gluteus minimus (collectively known as the "glute muscles"), as well as the biceps femoris (hamstrings) and core abdominal muscles. The classification of each activity considers not only the primary muscles involved but also the functional purpose of the movement and its application context in training scenarios, thereby providing a more comprehensive framework for evaluating the quality of movements generated by models.

To ensure a comprehensive dataset for evaluating human motion in text-to-video generation, we developed a structured data collection and description process, as shown in Figure 2a. Our approach emphasizes the diversity of movement types, clarity of video quality, and accuracy of motion de-

scriptions. This section outlines our methods for collecting and organizing video data, along with the steps taken to generate high-quality descriptions that accurately reflect each recorded action.

**Description Collection.** We use a multi-stage strategy to collect detailed descriptions for each video. The process involves the following steps:

*Action Identification.* We use Gemini-2.5 pro to locate each complete action accurately—instances containing multiple body parts—in the video recordings and label them with the appropriate event tags. During this stage, we discard all incomplete actions, such as those containing interruptions. And then, the Gemini-2.5 Pro model generates a series of candidate descriptions for each qualified video, capturing both the overall action flow and fine-grained motion details. To further refine these descriptions into concise and effective video prompts, we employ GPT-4o to rewrite them by aligning the textual content with the actual video context. This two-stage process ensures that the final prompts are both semantically faithful to the videos and directly usable for downstream text-to-video generation tasks.

*Description Validation.* Our team manually reviewed and corrected any inaccuracies, ambiguities, or incomplete descriptions, paying special attention to unclear action orientations or imprecise movement details. This validation process ensured that each description was both accurate and distinctive enough to properly identify the specific movement being performed.

## 4 MOVO BENCHMARKING METRICS

We propose three complementary metrics to comprehensively evaluate the similarity between motion sequences: Joint Angle Change (JAC), Dynamic Time Warping Similarity (DTW), and Motion Consistency Metric (MCM). These metrics are designed to capture different aspects of motion similarity, from low-level joint dynamics to high-level semantic consistency. A pose estimation model (Insafutdinov et al., 2016; Zhang et al., 2019; Jiang et al., 2023) is used to obtain the skeletal keypoints and joint features required for these metrics, ensuring accurate representation of human motion across frames.

**Joint Angle Change (JAC).** To capture joint articulation across frames, we define the Joint Angle Change (JAC) metric. For each frame $t$, the angle $\theta$ between selected joint vectors $\vec{v}_1$ and $\vec{v}_2$ (e.g., upper arm and forearm) is calculated as:

$$\bar{\theta} = \frac{1}{T} \sum_{t=1}^{T} \left( \frac{1}{N} \sum_{i=1}^{N} \arccos \left( \frac{\vec{v}_{i,1} \cdot \vec{v}_{i,2}}{\|\vec{v}_{i,1}\|\|\vec{v}_{i,2}\|} \right) \right) \tag{1}$$

where $T$ is the total number of frames in the video, $N$ is the total number of joint pairs for angle calculation, $\vec{v}_{i,1}$ and $\vec{v}_{i,2}$ are vectors representing the joint pair $i$, $\cdot$ denotes the dot product, and $\|\cdot\|$ represents the vector magnitude. To ensure consistency across frames, we calculate each joint's relative position $\vec{r}_{i,t}$ with respect to a reference joint (e.g., the hip) as:

$$\sigma_{\text{pos}} = \frac{1}{N} \sum_{i=1}^{N} \text{Var} \left( \{\vec{p}_{i,t} - \vec{p}_{\text{ref},t} \mid t = 1, \ldots, T\} \right) \tag{2}$$

where $\vec{p}_{i,t}$ is the position of joint $i$ at frame $t$, $\vec{p}_{\text{ref},t}$ is the position of the reference joint at frame $t$, $\text{Var}(\cdot)$ denotes the variance operation over all frames. For two videos, we calculate the Euclidean distance between their mean angle changes $\Delta\theta = |\bar{\theta}_1 - \bar{\theta}_2|$, where $\bar{\theta}_1$ and $\bar{\theta}_2$ are the mean angle changes of the two videos, and position variances $\Delta\sigma = |\sigma_{\text{pos},1} - \sigma_{\text{pos},2}|$, where $\sigma_{\text{pos},1}$ and $\sigma_{\text{pos},2}$ are the mean position variances of the two videos:

$$\text{distance} = \sqrt{(\Delta\theta)^2 + (\Delta\sigma)^2} \tag{3}$$

Finally, the similarity score JAC is normalized to the range $[0, 1]$ to indicate action similarity:

$$\text{JAC} = 1 - \frac{\text{distance}}{\text{max\_distance}} \tag{4}$$

where max_distance is a threshold indicating complete dissimilarity. This normalization provides an intuitive similarity metric, with higher scores indicating closer action resemblance.

**Dynamic Time Warping Similarity (DTW).** To quantify the similarity between the movements in two videos, we compute the Dynamic Time Warping distance between their skeletal keypoint sequences. For each video frame $t$, the positions of skeletal keypoints are extracted and represented as vectors $\vec{k}_t$. We then compute the relative change in keypoints across consecutive frames to capture motion dynamics:

$$\Delta \vec{k}_t = \vec{k}_t - \vec{k}_{t-1} \tag{5}$$

where $\Delta \vec{k}_t$ is the relative feature representing motion between frames $t$ and $t-1$. This process is repeated for all frames in each video to obtain a sequence of motion dynamics. Next, we flatten each frame's relative feature vector into a one-dimensional representation to facilitate distance computation. For a video with $T$ frames, the feature vector for each frame $t$ is defined as:

$$\text{flattened}_t = \text{flatten}(\Delta \vec{k}_t) \tag{6}$$

where $\text{flatten}(\cdot)$ denotes the operation of reshaping the vector into one dimension. To compute the similarity between two videos, we apply Dynamic Time Warping to measure the alignment cost between their sequences of flattened vectors. Given two videos with frame sequences $\{\text{flattened}_{1,t}\}_{t=1}^{T_1}$ and $\{\text{flattened}_{2,t}\}_{t=1}^{T_2}$, the DTW distance $D$ is calculated as:

$$D = \sum_{(t_1, t_2) \in \text{Path}} d(\text{flattened}_{1,t_1}, \text{flattened}_{2,t_2}) \tag{7}$$

where Path is the optimal alignment path minimizing cumulative Euclidean distance, and $d(\cdot, \cdot)$ denotes the Euclidean distance between two frames' flattened vectors.

Finally, to obtain a similarity score $S$, we normalize $D$ with a maximum allowable distance max_distance, ensuring the score falls between 0 and 1:

$$DTW = 1 - \frac{D}{\text{max\_distance}} \tag{8}$$

where $DTW$ represents the degree of similarity between the two videos, with higher values indicating greater alignment of movements.

**Motion Consistency Metric (MCM).** To assess whether two videos exhibit the same motion, we leverage a multi-modal large language model (MLLM) as a judge. The MLLM evaluates the videos and outputs a categorical result, indicating either "similar" or "not similar" based on the consistency of movements between the two videos (see Supplementary Materials for detailed prompt design).

The Motion Consistency Metric $MCM$ is defined as:

$$MCM = \begin{cases} 1, & \text{if MLLM outputs "similar"} \\ 0, & \text{if MLLM outputs "not similar"} \end{cases}$$

where $MCM$ yields a binary score representing the consistency of motion, with $MCM = 1$ indicating similar motions and $MCM = 0$ indicating dissimilar motions between the videos.

## 5 HUMAN VALIDATION

We conduct extensive human preference labeling on generated videos to validate whether our evaluation metrics align with human perception. Our annotation process follows a systematic pairwise comparison approach.

**Data Preparation.** For each movement type in our dataset, we generate videos using four different models: CogVideo, SVD, Open-Sora-Plan, Kling and compose them into groups. Specifically, given a text description $p_i$ describing a particular movement, we collect ten groups of Movement List videos, as shown in Table 3. Each group contains four videos generated by different models: $V_A, V_B, V_C, V_D$, where A,B,C,D represent different models.

**Pairwise Comparison.** Within each group, we create all possible pairs of videos for comparison. Given $M$ models, the number of pairs for each group is $\binom{M}{2} = \frac{M(M-1)}{2}$. In our case with $M = 4$, this results in six pairs: $(V_A, V_B), (V_A, V_C), (V_A, V_D), (V_B, V_C), (V_B, V_D), (V_C, V_D)$. The order of videos within each pair is randomized to prevent potential bias. For a prompt suite of $N$ text descriptions, this setup produces $N \times 10 \times \binom{4}{2} = 60N$ pairwise comparisons in total.

**Annotation Process.** Human annotators are asked to evaluate each video pair based on the realism of motion generation. For each comparison, annotators indicate their preference between the two videos. We ensure each pair receives ratings from multiple annotators to enhance reliability. The collected preferences are used to compute win ratios for each model and validate the alignment between our automated metrics and human perception.

**Win Ratio.** Based on human labels, we compute the win rate for each model through pairwise comparisons. The superior model received 1 point, the inferior model received 0 points, and in the case of a tie, both models received 0.5 points. Each model's win rate was calculated as the total score divided by the total number of pairwise comparisons it participated in, as detailed at Figure 6.

## 6 EXPERIMENT SETUP

### 6.1 MODELS

We selected 14 exemplary T2V models for evaluation, including both open-source and propriety models, including CogVideo (Hong et al., 2022), SD3+SVD (Blattmann et al., 2023a), Open-Sora-Plan (PKU-Yuan Lab and Tuzhan AI et al., 2024), Zeroscope (cerspense, 2023), Gen2 (Runway Research, 2023), Dream Machine (Luma AI, 2024), Kling (Kuaishou Technology, 2024), Pika 1.5 (Pika, 2024),Wan 2.1 (Team Wan et al., 2025),

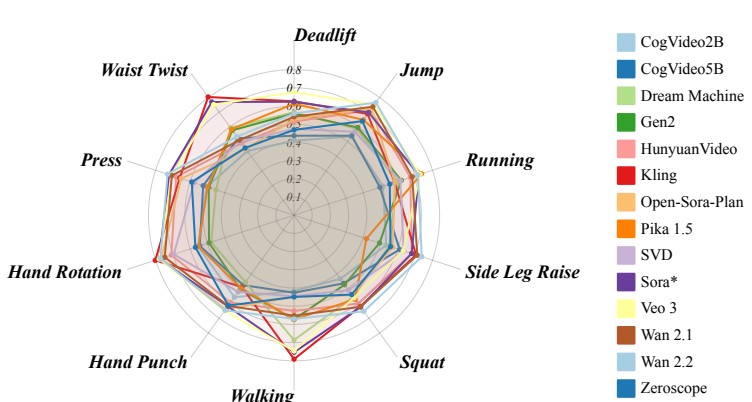

Figure 3: Average of JAC, DTW, and MCM for lower and upper body movements (excluding Sora due to limited evaluation data).

Wan 2.2 (Wan-Video Team, 2025), Veo 3 (Google DeepMind, 2025), HunyuanVideo (Kong et al., 2025) and Sora (OpenAI, 2024). For more detailed, please refer to the Supplementary Materials.

### 6.2 EXPERIMENT DESIGN

In this experiment, we used the prompts from the Posture Dataset for inference on 14 tested models. Each model generated 893 videos. Subsequently, using the metrics defined in Section 4, the generated videos were compared with the videos in the Posture Dataset (Ground Truth) to compute the evaluation metrics. Due to OpenAI's restrictions on Sora, only 10 randomly selected prompts

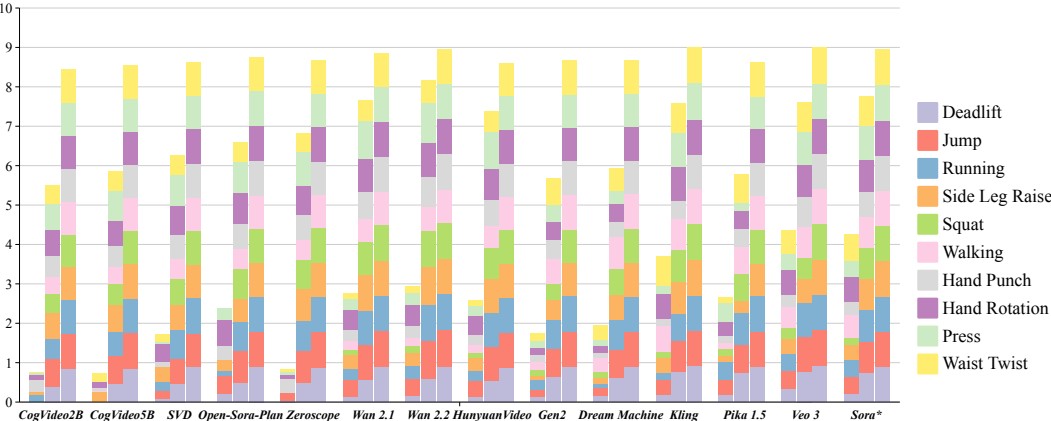

Figure 4: An overview of the evaluation results across all models. This figure summarizes 14 T2V models, where each model forms a group of three stacked bars (JAC, DTW and MCM) and the stack segments correspond to the 10 actions. The bar height equals to the sum of normalized scores when higher is better. Models are arranged from open-source to proprietary, and Sora* is reported with limited data. The plot makes it easy to see per-model trade-offs and where strengths concentrate by action family.

per category were used for video generation, making the evaluation results preliminary and for reference only. For Veo 3, we accessed the model via the official API (self-hosting unavailable), and generations reflect the API's default settings at evaluation time.

# 7 EVALUATION RESULTS

We employed YOLO-X (Gillani et al., 2022) to detect humans in the videos, feeding the detected regions into the RTMPose-X (Jiang et al., 2023) model to extract skeletal structures and keypoint information. For evaluation, we compared the skeletal structures in the generated videos to those in our dataset videos, which served as Ground Truth. This comparison was based on keypoint coordinates for each frame, enabling us to compute metrics that evaluate the quality of the generated videos and their similarity to real-world videos, as shown in Figure 2b. If the prompt for generating the video includes "hand," we applied the RTMPose-M simcc hand5 (Jiang et al., 2023) model to specifically extract skeletal structures and keypoints for the hands. This allows for a more granular analysis of hand movements, enhancing the precision of our evaluation metrics for videos with a focus on hand gestures or actions. We computed the unnormalized maximum distances for the JAC and DTW metrics and set max_distance to 1000. For all open-source models, we set the seed parameter to 88, while keeping all other hyperparameters at their default values. The results are shown in Figure 4. For more detailed results, such as experiments on more complex motions and pose estimation models, please refer to Appendix F.

## 7.1 JAC EVALUATION ON MOVO

Table 4 reports joint-articulation consistency (JAC). We observe strong intra-model variability across actions: models that score well on upper-body tasks often drop on lower-body control. For instance, Open-Sora-Plan reaches 0.371 on *hand punch* yet shows weaker articulation on legs. Pika 1.5 illustrates the gap when it gains 0.467 on *running* but 0.145 on *side leg raise*. *Sora* is comparatively balanced: moderate on *deadlift* and *squat*, and stronger on continuous lower-body motions, with mixed results on faster upper-body actions. Current models capture gross motion classes but struggle with fine-grained joint articulation, especially for lower limbs requiring precise coordination.

## 7.2 DTW EVALUATION ON MOVO

Table 5 evaluates temporal alignment via dynamic time warping similarity (DTW). Proprietary models (Kling 1.0, Pika 1.5) show strong alignment on complex actions, yet consistency is not universal: Pika 1.5 performs well on *walking* with a score of 0.701 but drops to 0.300 on *side leg raise*, indicating difficulty with isolated or abrupt motions. *Sora* maintains comparatively even alignment across

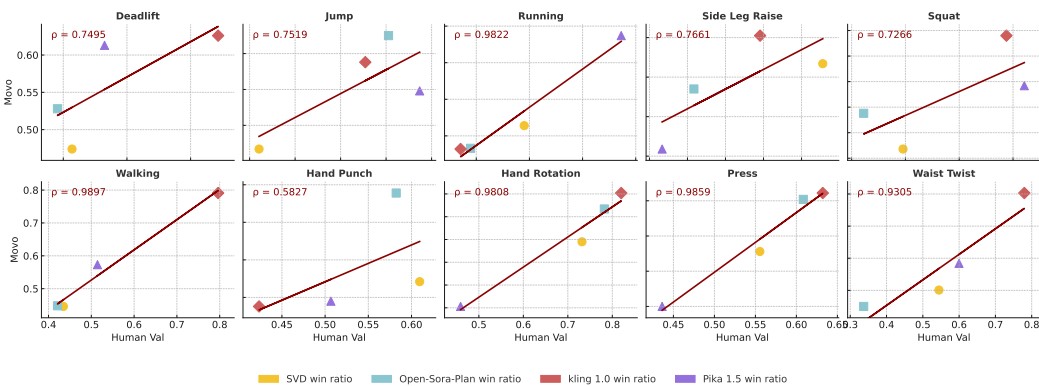

Figure 5: Correlation of Movo Evaluation (Average of JAC, DTW and MCM Metrics) with Human Annotations Across Different Human Motion Types

both dynamic and controlled actions. In all, Flow-like continuity is easier to achieve in steady periodic movements than in actions with discrete phases or brief holds.

## 7.3 MCM EVALUATION ON MOVO

Table 6 reports structural consistency using the Motion Consistency Metric (MCM). In general, Kling 1.0 leads on most movements. Among open-source baselines, Open-Sora-Plan and Zeroscope are competitive on select classes. *Sora* is uniformly strong, with scores tightly clustered around 0.88–0.90 across both lower- and upper-body actions, suggesting robust preservation of overall motion structure. MCM also reveals weaknesses in nuanced upper-body control. Moreover, the binary nature of MCM can mask subtle fidelity gaps even when structures look similar. Overall, preserving coarse structure is increasingly reliable, but capturing fine-grained coherence remains challenging, motivating joint- and phase-aware diagnostics.

## 7.4 VALIDATING HUMAN ALIGNMENT OF MOVO

Human scores were calculated the models' win rates over 1200 comparisons (N=2), providing a robust dataset to evaluate these correlations. For each type of human motion, we based on Movo's evaluation results (Average of JAC, DTW and MCM Metrics) and human scores results, as shown in Figure 5. The human scores for different models are displayed across various motion categories. In each figure, we observe the correlation coefficient $\rho$ between Movo's metrics and human evaluations, such as 0.9859 in Hand Punch and 0.9897 in Walking. Notably, high correlations are observed in motions like Running ($\rho = 0.9822$), Walking ($\rho = 0.9897$), Hand Rotation ($\rho = 0.9808$), and Press ($\rho = 0.9859$). The results reveal an overall high consistency between automated evaluation scores and human annotations, with average correlation values supporting the validity of Movo as a metric.

## 8 CONCLUSION

Based on the evaluation metrics and experimental results presented, we derive the following key insights: *(1) Performance varies by motion type.* Lower-body actions score higher on JAC/DTW/MCM than upper-body actions. *Sora* is comparatively balanced across both groups in Fig. 3. *(2) Non-uniformity and bias across models.* Proprietary systems generally outperform open-source baselines, but gains concentrate on upper-body tasks under MCM, suggesting specialization rather than robustness in Table 4 and Table 5. *Sora* shows more even performance despite limited accessible data. *(3) Missing fine-grained dynamics.* Open-source models often fail to capture subtle joint articulation; DTW exposes rhythm drift even when videos appear smooth. *Sora* is not exempt.

We present **Movo**, a kinematics-centric benchmark for human-motion realism in T2V. Movo couples posture-focused, camera-aware prompts with three skeletal metrics to yield interpretable, body-centric scores. Evaluating a representative set of leading open and proprietary models, Movo exposes persistent gaps in biomechanical plausibility and temporal consistency, providing actionable diagnostics for model selection, quality gating, and future research.

ETHICS STATEMENT

This work focuses on evaluating human motion realism in text-to-video (T2V) generation. All video data collected in the posture dataset were either sourced from public platforms with permissive licenses or recorded with informed consent from participants. Before recording, volunteers were shown instructional materials and provided written consent, with the option to withdraw at any time. Personally identifiable information was excluded, and only body movements relevant to evaluation were retained. We acknowledge that T2V systems pose potential ethical and societal risks, including the generation of misleading or unsafe human motions. Implausible motions may encourage viewers, particularly juveniles, to imitate harmful behaviors, while synthetic videos can also be misused for disinformation or unauthorized likeness replication. Our benchmark does not generate or distribute harmful content; rather, it aims to surface biomechanical errors and promote safer, more realistic human motion generation. By releasing Movo, we intend to provide the community with tools to improve the safety, reliability, and transparency of T2V models. We encourage responsible use of our dataset and benchmark, and we explicitly discourage applications that could compromise human well-being, propagate misinformation, or violate privacy or likeness rights.

REPRODUCIBILITY STATEMENT

To promote transparency and reproducibility, we release all the code and scripts accompanying this paper in the Supplementary Materials.

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

## A   SIMPLIFICATION OF MOTION TAXONOMY

To ensure a clear and practical classification, we categorized human activities based on the primary body parts involved. While this taxonomy simplifies complex human motions, it remains effective for analyzing movements that significantly influence joint positions and biomechanical dynamics. Below, we elaborate on the rationale for our choices and the exclusions.

**Exclusion of Facial Movements**   Facial movements, while important in human communication and emotional expression, were excluded from this taxonomy. This decision was made because facial motions primarily involve micro-expressions and small-scale muscular changes, which are insufficient to produce measurable joint displacement or contribute to broader body kinematics.

**Focus on Major Muscle Groups**   The taxonomy divides movements into upper and lower body activities, which aligns with the natural grouping of muscle synergies in physical activities. Although some exercises, like deadlifts, engage the entire body, they are categorized under lower body movements due to the dominant involvement of leg and hip muscles. For similar reasons, activities such as pull-ups, while engaging the upper body extensively, could also be conceptually grouped under "deadlift" due to overlapping muscle recruitment patterns. However, for simplicity, we kept them distinct under the upper body classification to emphasize specificity.

**Simplification for Practicality**   While the human body contains many fine-grained muscle groups, analyzing activities at such granularity adds complexity without significant benefits in typical motion analysis applications. Thus, we opted for broader categories that better align with real-world activities and the synergistic functions of muscle groups. For example: *1) Upper Body Movements:* This category includes activities such as pressing and hand rotation, which highlight the dominant role of the shoulders and arms. *2) Lower Body Movements:* Activities such as squats and jumping focus on the legs and hips as primary movers.

**Exclusion of Other Specialized Movements**   Movements involving smaller muscle groups (e.g., fingers, toes) or specialized actions (e.g., fine motor skills) were excluded. These activities have minimal impact on joint displacement and are less relevant to the core physical activities that this taxonomy aims to address.

**Upper Body Inclusion of Compound Movements**   Compound movements like deadlifts or pull-ups were considered for their overlap between upper and lower body categories. For example, deadlifts, though categorized under lower body activities, involve substantial engagement of the upper body, such as grip strength and spinal stabilization. These nuances were carefully accounted for while simplifying the taxonomy.

This streamlined taxonomy ensures that the classification is easy to interpret, aligns with kinesiological principles, and remains relevant for most applications, from biomechanics research to physical activity monitoring.

## B   MLLMS FOR VIDEO DESCRIPTION

The task of generating accurate and detailed video descriptions is critical for applications ranging from video retrieval to content analysis and accessibility enhancement. Multimodal large language models (MLLMs) have emerged as powerful tools for this task by combining visual and textual modalities to produce coherent and informative descriptions. This section discusses the role of MLLMs in video description tasks and introduces a set of structured prompts designed to guide the models' outputs effectively.

**Role of Prompts in Video Description**   Prompts play a pivotal role in shaping the responses of MLLMs, particularly in complex tasks like video description. A well-designed prompt can guide the model to focus on specific aspects of the video content, ensuring that the generated descriptions are not only accurate but also relevant to the intended application. For this purpose, we created a set of 10 prompts tailored to elicit detailed, action-oriented descriptions while avoiding unnecessary or biased information (see Table 1).

**Objectives of Prompt Design**   The prompts in Table 1 are carefully crafted to achieve the following objectives: 1. Focus on Actions and Events: Each prompt emphasizes the actions and sequences occurring in the video, ensuring that the descriptions remain centered on the core content. 2. Inclu-

Table 1: Prompts for video description tasks

| ID | Prompt |
|----|--------|
| 1 | Describe this video focusing on the actions being performed. Where is the camera positioned? Ignore the gender of the people in the video. |
| 2 | Explain what is happening in the video with an emphasis on the sequence of actions and their purpose. Camera details like angles and movement are important. |
| 3 | Provide a detailed description of the video content, focusing only on the actions and camera positioning. Avoid mentioning any physical appearances. |
| 4 | What activities are being performed in the video? Mention the camera's perspective and movement, while ignoring the subjects' identity. |
| 5 | Focus on describing the events and actions in the video. Where is the camera placed, and what angles are used? Do not include details about the participants' gender or appearance. |
| 6 | Summarize the video by explaining the actions taking place. Note the camera's position and transitions, but do not consider any personal attributes of the people involved. |
| 7 | Identify the key actions occurring in this video. Emphasize the camera's role in capturing the actions, excluding personal details of the individuals. |
| 8 | Analyze the video for the activities being shown. Pay attention to camera angles and positioning while disregarding the participants' physical descriptions. |
| 9 | What movements and actions are captured in this video? Highlight the camera's perspective, avoiding any focus on the individuals' appearance or gender. |
| 10 | Describe the sequence of actions in this video, focusing on the activities and the camera's placement. Avoid any mention of the participants' personal characteristics. |

Table 2: Comparison of Movo with widely used T2V benchmarks

| Benchmark | Kinematics | Contact/Phys. | Temporal | Camera Ctrl. | Human Eval. |
|-----------|-----------|---------------|----------|--------------|-------------|
| VBench | ✗ | ✗ | △ | △ | △ |
| EvalCrafter | ✗ | ✗ | △ | ✗ | △ |
| T2V-CompBench | ✗ | ✗ | ✗ | △ | △ |
| Video-Bench | ✗ | ✗ | △ | △ | △ |
| PhyGenBench | ✗ | ✓ | △ | ✗ | △ |
| **Movo (ours)** | ✓ | ✓ | ✓ | ✓ | ✓ |

*Legend:* ✓ explicitly covered; △ indirect or limited coverage; ✗ not covered.

sion of Camera Details: Understanding the role of the camera in capturing video content, such as its placement, movement, and perspective, is crucial. The prompts explicitly encourage the model to include these aspects. 3. Exclusion of Personal Attributes: To ensure objectivity and ethical use, the prompts explicitly instruct the model to avoid describing personal characteristics such as the gender or appearance of individuals in the video. This mitigates potential biases and ensures privacy.

**Application Scenarios** The prompts were designed to cater to a wide range of video types, including: 1. Instructional Videos: Where sequences of actions and their purpose are central to the description. 2. Surveillance Footage: Where camera positioning and actions captured are crucial for analysis. 3. Sports and Performance: Where the emphasis is on the movements and activities performed.

**Model Selection and Implementation** Finally, we selected the state-of-the-art model, `Qwen2-vl` (Wang et al., 2024a), to describe our collected text-video dataset. For each video, a random prompt from the ten provided in Table 1 was used to ensure diverse and context-appropriate descriptions.

## C  HUMAN ANNOTATION

In this study, we employed a rigorous human annotation process to evaluate the effectiveness of video content in matching given tags. Ten PhD student volunteers, comprising an equal distribution of five male and five female participants, were selected to conduct the annotations. The participants were trained in video analysis to ensure consistent and accurate evaluations.

For the annotation process, the volunteers were presented with pairs of videos, as shown in the figure, along with a corresponding tag such as "Boxing." Their task was to determine which video better matched the tag based on the visual and contextual content of the videos. Each pair of videos

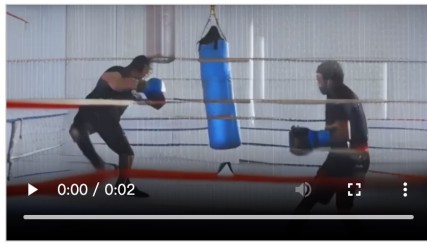

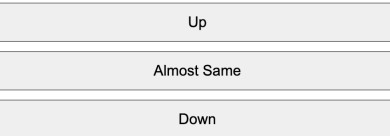

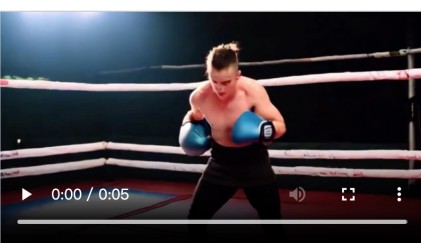

Figure 6: Annotation interface for video evaluation: Annotators compare two video clips with the tag 'Boxing' and select the better match using options 'Up,' 'Almost Same,' or 'Down.

was displayed alongside three options for evaluation: "Up" (indicating the top video matches better), "Down" (indicating the bottom video matches better), or "Almost Same" (indicating both videos are equally relevant), as shown in Figure 6.

The annotation interface was designed to minimize cognitive load and maximize accuracy by providing a clear layout and intuitive options. The volunteers were instructed to carefully consider the movements, settings, and actions depicted in each video before making their decisions. Each annotation task was independently performed by all ten participants to ensure diversity in perspectives and reduce bias.

The collected annotations were aggregated and analyzed to measure inter-annotator agreement, providing a reliable foundation for assessing the quality of the videos in relation to their tags. This human-centered evaluation approach contributed significantly to validating the results of our study.

Our hiring criteria for manual verification are:

1) A 45-minute training session covering common motion failures produced by current T2V models, such as missing fingers, duplicated limbs, joint misplacement, unrealistic bone structure, and inconsistent arm–leg articulation.

2) Set a quiz about a calibration exam of 30 videos, requiring ¿=90% agreement with verified answers before annotation.

3) Clear category-specific guidelines for valid vs. invalid actions, with visual example

## D    DATASET VISUALIZATION

The dataset visualization aims to provide an overview of the ground truth data used for human motion analysis. Figure 7 presents videos depicting different exercises with overlaid skeletal keypoints. These keypoints represent the critical joints and body parts tracked during the movements, offering a detailed view of pose estimation and motion tracking accuracy.

The visualizations include a variety of motion. Each activity is captured across multiple frames to demonstrate the temporal progression of the actions. The skeletal keypoints are color-coded and connected to highlight joint positions and limb orientations, enabling clear interpretation of the body's posture and motion dynamics.

This visualization helps to validate the quality of the dataset by showcasing its ability to capture diverse human motions with high precision. The overlaid skeletons indicate that the pose estimation aligns well with the physical movements depicted in the images, supporting its application in motion

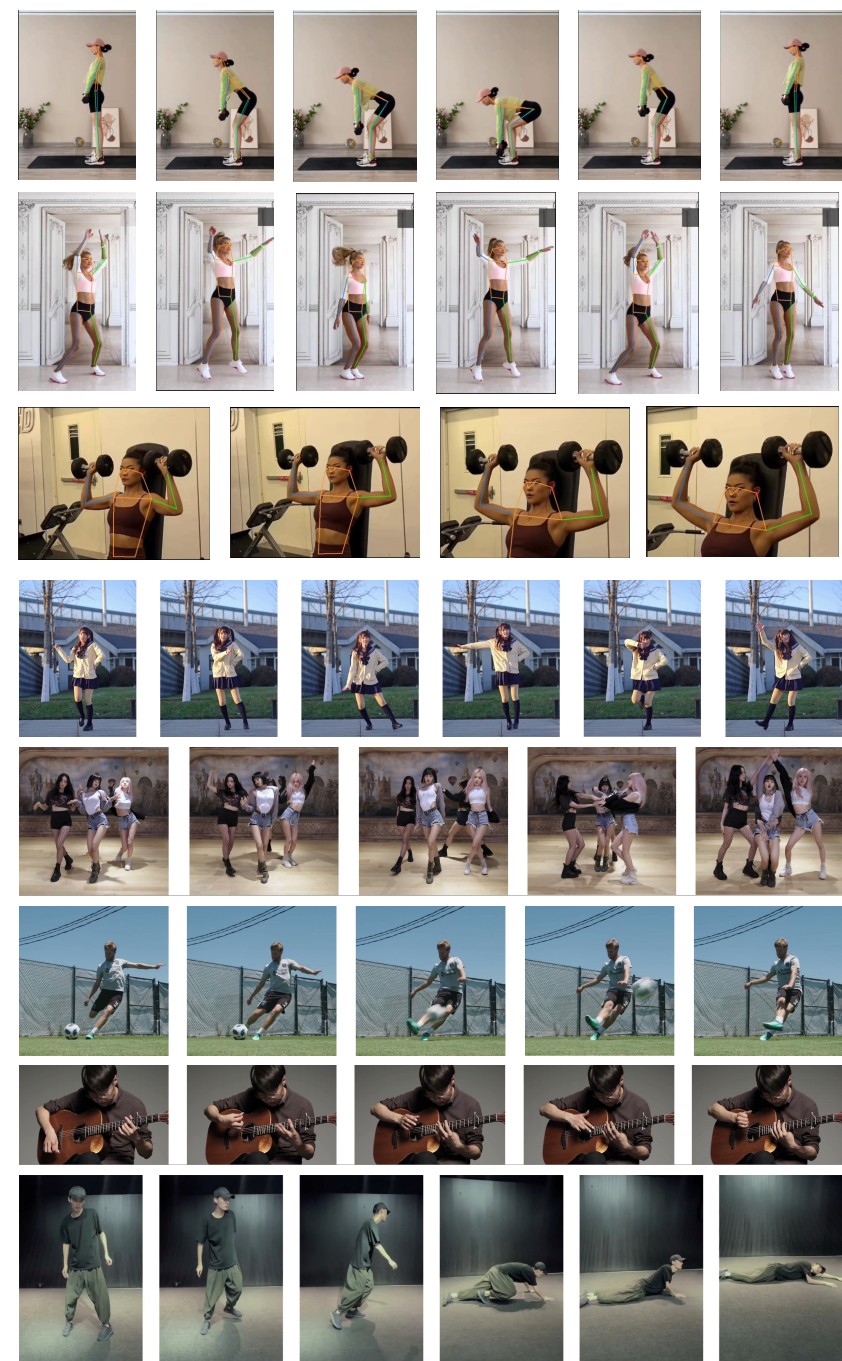

Figure 8: Visualization of motion analysis scenes.

analysis tasks. Furthermore, the variety in activities underscores the dataset's comprehensiveness and versatility for studying a broad range of human actions.

## E   EXTENDED RELATED WORK

The latest breakthroughs in generative AI, particularly with the development of Transformer models (Hong et al., 2022; Villegas et al., 2022; Wu et al., 2021; 2022; Gupta et al., 2022; Yu et al., 2023) and diffusion models (Ho et al., 2022a;b; Blattmann et al., 2023b; He et al., 2022; Khachatryan et al., 2023; Luo et al., 2023; Singer et al., 2022; Wang et al., 2023f; Sun et al., 2024), have significantly advanced open-domain video generation. Transformer-based approaches encode videos as discrete

Table 3: Movement classification

| Category | Movement list |
|---|---|
| Lower body movements | Deadlift; Jump; Running; Side leg raise; Squat; Walking |
| Upper body movements | Hand punch; Hand rotation; Press; Waist twist |

Table 4: Lower and Upper Body Movements Evaluation Using JAC Metric (* limited data)

| Model | Lower Body Movements | | | | | | Upper Body Movements | | | |
|---|---|---|---|---|---|---|---|---|---|---|
| | Deadlift | Jump | Running | Side Leg Raise | Squat | Walking | Hand Punch | Hand Rotation | Press | Waist Twist |
| *Open-source Models* | | | | | | | | | | |
| CogVideo2B | 0 | 0 | 0.170 | 0.097 | 0 | 0 | 0.306 | 0.138 | 0.027 | 0.008 |
| CogVideo5B | 0 | 0 | 0 | 0.277 | 0 | 0.006 | 0.077 | 0.147 | 0 | 0.224 |
| SVD | 0.083 | 0.207 | 0.213 | 0.401 | 0 | 0 | 0.105 | 0.476 | 0.061 | 0.180 |
| Open-Sora-Plan | 0.197 | **0.479** | 0.135 | 0.257 | 0 | 0 | **0.371** | 0.649 | 0.285 | 0 |
| Zeroscope | 0.028 | 0.211 | 0 | 0 | 0 | 0 | 0.360 | 0.103 | 0.065 | 0.051 |
| Wan 2.1 | 0.152 | 0.410 | 0.295 | 0.338 | 0.142 | 0.211 | 0.284 | 0.512 | 0.278 | 0.143 |
| Wan 2.2 | 0.163 | 0.432 | 0.311 | 0.352 | 0.157 | 0.227 | 0.297 | 0.539 | 0.293 | 0.158 |
| HunyuanVideo | 0.141 | 0.384 | 0.276 | 0.319 | 0.132 | 0.198 | 0.261 | 0.481 | 0.254 | 0.131 |
| *Proprietary Models* | | | | | | | | | | |
| Gen2 | 0.136 | 0.179 | 0.243 | 0.113 | 0.158 | 0.191 | 0.189 | 0.172 | 0.193 | 0.179 |
| Dream Machine | 0.167 | 0.191 | 0.118 | 0.158 | 0.129 | 0.362 | 0.142 | 0.154 | 0.172 | 0.362 |
| Kling | **0.197** | 0.370 | 0.169 | **0.401** | 0.138 | **0.673** | 0.156 | **0.649** | 0.198 | **0.761** |
| Pika 1.5 | 0.192 | 0.374 | **0.467** | 0.145 | **0.182** | 0.138 | 0.177 | 0.374 | **0.467** | 0.148 |
| Veo 3 | 0.344 | 0.445 | 0.432 | 0.391 | 0.264 | 0.528 | 0.323 | 0.621 | 0.406 | 0.598 |
| Sora* | 0.219 | 0.422 | 0.438 | 0.382 | 0.179 | 0.584 | 0.338 | 0.612 | 0.414 | 0.682 |

visual tokens, which are then generated automatically (Yuan et al., 2024; Liu et al., 2024b). On the other hand, diffusion models have been widely explored for this task to reduce the high computational cost of video generation, demonstrating superior capabilities (Ho et al., 2022a;b; Blattmann et al., 2023b).

Diffusion models, such as Make-A-Video (Singer et al., 2022), leverage pre-trained image diffusion models and enhance their video generation capabilities by fine-tuning temporal attention mechanisms. VideoLDM (Blattmann et al., 2023b) introduces a multi-stage alignment process in latent space to generate high-resolution videos. Similarly, GEST (Masala et al., 2023) employs graph-based representations to encode the spatio-temporal relationships between text and video, generating contextually rich content.

To enhance controllability, methods such as VideoComposer (Wang et al., 2024b) incorporate additional guidance signals, such as depth maps, ensuring that the generated videos align more closely with textual prompts. Meanwhile, VideoDirectorGPT (Lin et al., 2023) leverages GPT-4 (Achiam et al., 2023) to create scene layouts and control specific video compositions. Other approaches, such as Tune-A-Video (Wu et al., 2023), implement temporal self-attention modules in pre-trained diffusion models, achieving higher fidelity in text-driven video generation.

The introduction of diffusion transformers (Peebles & Xie, 2023; Bao et al., 2023; Gao et al., 2023) has further revolutionized video generation, leading to advanced methods like Latte (Ma et al., 2024) and Sora (OpenAI, 2024). These methods have been applied in various domains.

## F  ADDITIONAL EXPERIMENTS

In this section, we address the constructive feedback provided by reviewers regarding dataset diversity, metric robustness, and evaluation fairness. We have significantly expanded the benchmark with new challenge categories and conducted rigorous ablation studies to validate the stability and instructional value of our metrics.

### F.1  EXPANSION OF DATASET: CHALLENGE CATEGORIES

To resolve that the original Movo dataset, while a robust starting point, was limited to structured fitness motions and might not represent the full complexity of human movement. We appreciate this

Table 5: Lower and Upper Body Movements Evaluation Using DTW Metric (* limited data)

| Model | Lower Body Movements | | | | | | Upper Body Movements | | | |
|---|---|---|---|---|---|---|---|---|---|---|
| | Deadlift | Jump | Running | Side Leg Raise | Squat | Walking | Hand Punch | Hand Rotation | Press | Waist Twist |
| *Open-source Models* | | | | | | | | | | |
| CogVideo2B | 0.381 | 0.724 | 0.513 | 0.663 | 0.465 | 0.431 | 0.524 | 0.678 | 0.667 | 0.461 |
| CogVideo5B | 0.451 | 0.730 | 0.608 | 0.684 | 0.538 | 0.441 | 0.508 | 0.637 | 0.754 | 0.494 |
| SVD | 0.459 | 0.634 | 0.739 | 0.642 | 0.666 | 0.498 | 0.598 | 0.729 | 0.812 | 0.483 |
| Open-Sora-Plan | 0.497 | 0.797 | 0.734 | 0.594 | 0.762 | 0.503 | **0.655** | 0.762 | 0.802 | 0.499 |
| Zeroscope | 0.498 | **0.805** | 0.770 | 0.793 | 0.747 | 0.516 | 0.623 | 0.737 | 0.847 | 0.480 |
| Wan 2.1 | 0.572 | 0.892 | 0.853 | 0.909 | 0.834 | 0.596 | 0.685 | 0.839 | 0.959 | 0.528 |
| Wan 2.2 | 0.603 | 0.944 | 0.927 | 0.961 | 0.902 | 0.624 | 0.751 | 0.877 | 1.009 | 0.574 |
| HunyuanVideo | 0.532 | 0.870 | 0.861 | 0.852 | 0.808 | 0.549 | 0.669 | 0.787 | 0.939 | 0.509 |
| *Proprietary Models* | | | | | | | | | | |
| Gen2 | 0.641 | 0.719 | 0.717 | 0.520 | 0.418 | 0.637 | 0.464 | 0.452 | 0.446 | 0.681 |
| Dream Machine | 0.632 | 0.689 | 0.773 | 0.630 | 0.673 | 0.797 | 0.384 | 0.444 | 0.351 | 0.561 |
| Kling | **0.770** | 0.794 | 0.686 | **0.803** | 0.812 | 0.800 | 0.457 | **0.847** | **0.866** | **0.747** |
| Pika 1.5 | 0.747 | 0.691 | **0.835** | 0.300 | 0.670 | 0.701 | 0.457 | 0.444 | 0.223 | 0.725 |
| Veo 3 | 0.764 | 0.899 | 0.851 | 0.611 | 0.529 | 0.800 | 0.744 | 0.827 | 0.830 | 0.736 |
| Sora* | 0.751 | 0.783 | 0.822 | 0.768 | 0.790 | 0.784 | 0.638 | 0.824 | 0.853 | 0.736 |

Table 6: Lower and Upper Body Movements Evaluation Using MCM Metric (* limited data)

| Model | Lower Body Movements | | | | | | Upper Body Movements | | | |
|---|---|---|---|---|---|---|---|---|---|---|
| | Deadlift | Jump | Running | Side Leg Raise | Squat | Walking | Hand Punch | Hand Rotation | Press | Waist Twist |
| *Open-source Models* | | | | | | | | | | |
| CogVideo2B | 0.85 | 0.88 | 0.86 | 0.84 | 0.83 | 0.82 | 0.84 | 0.85 | 0.82 | 0.84 |
| CogVideo5B | 0.86 | 0.89 | 0.88 | 0.87 | 0.85 | 0.83 | 0.84 | 0.85 | 0.82 | 0.85 |
| SVD | 0.88 | 0.86 | 0.89 | 0.86 | 0.86 | 0.84 | 0.86 | 0.88 | 0.86 | 0.84 |
| Open-Sora-Plan | 0.89 | **0.90** | 0.88 | 0.86 | 0.87 | 0.84 | **0.89** | 0.89 | 0.87 | 0.85 |
| Zeroscope | 0.88 | 0.90 | 0.89 | 0.88 | 0.87 | 0.83 | 0.86 | 0.87 | 0.86 | 0.84 |
| Wan 2.1 | 0.90 | 0.91 | 0.90 | 0.89 | 0.89 | 0.85 | 0.88 | 0.89 | 0.88 | 0.86 |
| Wan 2.2 | **0.91** | 0.92 | **0.91** | 0.90 | 0.90 | 0.86 | 0.89 | **0.90** | **0.89** | 0.87 |
| HunyuanVideo | 0.87 | 0.89 | 0.88 | 0.87 | 0.86 | 0.83 | 0.85 | 0.86 | 0.85 | 0.83 |
| *Proprietary Models* | | | | | | | | | | |
| Gen2 | 0.90 | 0.89 | 0.90 | 0.85 | 0.84 | 0.89 | 0.85 | 0.85 | 0.84 | 0.87 |
| Dream Machine | 0.90 | 0.88 | 0.90 | 0.86 | 0.86 | 0.90 | 0.84 | 0.84 | 0.83 | 0.86 |
| Kling | **0.91** | 0.90 | 0.89 | **0.91** | **0.91** | **0.90** | 0.85 | **0.91** | **0.92** | **0.90** |
| Pika 1.5 | 0.90 | 0.88 | **0.91** | 0.81 | 0.86 | 0.88 | 0.85 | 0.84 | 0.81 | 0.88 |
| Veo 3 | 0.92 | 0.91 | 0.90 | 0.89 | 0.89 | 0.91 | 0.88 | 0.89 | 0.88 | 0.92 |
| Sora* | 0.90 | 0.89 | 0.90 | 0.89 | 0.90 | 0.89 | 0.88 | 0.90 | 0.90 | 0.89 |

perspective and fully agree that broader coverage is valuable. While Movo was designed as a foundational benchmark for atomic actions, we acknowledge the need to test models on more chaotic scenarios.

To address this, we manually collected and annotated **486 additional videos** sourced from Motion-X and YouTube to represent four new "Challenge Categories." These categories were specifically selected to target the weaknesses identified:

- **Falling:** Represents non-periodic, physics-driven, and reactive motion where gravity and momentum are critical (simulating "slipping").

- **Ball Games:** Represents dynamic human-object interaction and hand-eye coordination.

- **Playing Instruments:** Represents fine-grained control and precise limb positioning.

- **Dance:** Represents high-degree-of-freedom (DoF) kinematics and diverse, non-standard poses.

We re-evaluated all models on this expanded benchmark. The results, presented in Tables 7, 8, and 9, reveal a significant performance drop compared to the original categories. For instance, even state-of-the-art models like Sora 2 and Veo 3 show a ∼30-50% drop in JAC scores on "Falling" compared to "Walking." This confirms that while models may master basic patterns, they struggle significantly with emergent, physics-based scenarios.

Table 7: Evaluation of "Challenge Categories" (Complex/Everyday Motion) Using JAC Metric

| Model | Challenge Categories | | | |
|---|---|---|---|---|
| | Falling | Ball Games | Instruments | Dance |
| ***Open-source Models*** | | | | |
| CogVideo2B | 0.004 | 0.015 | 0.038 | 0.091 |
| CogVideo5B | 0.013 | 0.031 | 0.067 | 0.121 |
| SVD | 0.045 | 0.084 | 0.119 | 0.149 |
| Open-Sora-Plan | 0.093 | 0.147 | 0.179 | 0.214 |
| Zeroscope | 0.017 | 0.027 | 0.059 | 0.089 |
| Wan 2.1 | 0.103 | 0.201 | 0.223 | 0.244 |
| Wan 2.2 | 0.128 | 0.213 | 0.236 | 0.268 |
| HunyuanVideo | 0.099 | 0.192 | 0.209 | 0.227 |
| ***Proprietary Models*** | | | | |
| Gen2 | 0.081 | 0.122 | 0.138 | 0.173 |
| Dream Machine | 0.088 | 0.109 | 0.131 | 0.152 |
| Kling | **0.152** | 0.238 | 0.298 | 0.322 |
| Pika 1.5 | 0.119 | 0.161 | 0.188 | 0.236 |
| Veo 3 | 0.214 | 0.298 | 0.331 | 0.401 |
| Sora 2 | **0.309** | **0.425** | **0.447** | **0.518** |

Table 8: Evaluation of "Challenge Categories" Using DTW Metric (Temporal Alignment)

| Model | Falling | Ball Games | Instruments | Dance |
|---|---|---|---|---|
| ***Open-source Models*** | | | | |
| CogVideo2B | 0.148 | 0.202 | 0.217 | 0.244 |
| CogVideo5B | 0.182 | 0.229 | 0.243 | 0.279 |
| SVD | 0.199 | 0.261 | 0.278 | 0.287 |
| Open-Sora-Plan | 0.221 | 0.276 | 0.288 | 0.318 |
| Zeroscope | 0.185 | 0.245 | 0.268 | 0.297 |
| Wan 2.1 | 0.251 | 0.302 | 0.344 | 0.365 |
| Wan 2.2 | 0.268 | 0.333 | 0.358 | 0.387 |
| HunyuanVideo | 0.229 | 0.306 | 0.321 | 0.349 |
| ***Proprietary Models*** | | | | |
| Gen2 | 0.218 | 0.256 | 0.284 | 0.301 |
| Dream Machine | 0.211 | 0.243 | 0.262 | 0.294 |
| Kling | **0.326** | 0.368 | 0.374 | 0.402 |
| Pika 1.5 | 0.243 | 0.292 | 0.314 | 0.343 |
| Veo 3 | 0.347 | 0.366 | 0.405 | 0.430 |
| Sora 2 | **0.364** | **0.398** | **0.423** | **0.439** |

## F.2 ROBUSTNESS OF METRICS AND ESTIMATORS

To prove that the benchmark did not relies on the RTMPose-X estimator, suggesting that generation artifacts might cause pose estimation errors that propagate into the scores. We fundamentally argue that estimator failure is a signal instead of a noise. In current T2V systems, pose extraction fails primarily when motion is physically implausible. For example, limb hallucinations cause estimator flickering, and blurred limbs break tracking. These artifacts directly reflect biomechanical implausibility, and our metrics explicitly measure this instability.

To empirically validate that our rankings are not biased by a specific estimator, we re-evaluated the benchmark using two alternative architectures: BlazePose (Bazarevsky et al., 2020) and YOLOv8l-pose (Jocher & Qiu, 2024). As shown in Table 10, the results demonstrate high agreement with our primary RTMPose-X evaluation, yielding a Spearman rank correlation of $\rho > 0.94$ for both JAC and

Table 9: Evaluation of "Challenge Categories" Using MCM Metric (Semantic Consistency)

| Model | Falling | Ball Games | Instruments | Dance |
|---|---|---|---|---|
| ***Open-source Models*** | | | | |
| CogVideo2B | 0.21 | 0.27 | 0.26 | 0.31 |
| CogVideo5B | 0.25 | 0.33 | 0.29 | 0.36 |
| SVD | 0.34 | 0.39 | 0.35 | 0.43 |
| Open-Sora-Plan | 0.38 | 0.44 | 0.41 | 0.48 |
| Zeroscope | 0.32 | 0.35 | 0.34 | 0.41 |
| Wan 2.1 | 0.47 | 0.55 | 0.52 | 0.59 |
| Wan 2.2 | 0.54 | 0.57 | 0.60 | 0.63 |
| HunyuanVideo | 0.43 | 0.50 | 0.46 | 0.56 |
| ***Proprietary Models*** | | | | |
| Gen2 | 0.38 | 0.47 | 0.42 | 0.49 |
| Dream Machine | 0.44 | 0.45 | 0.43 | 0.51 |
| Kling | 0.59 | 0.67 | 0.61 | 0.70 |
| Pika 1.5 | 0.46 | 0.52 | 0.49 | 0.57 |
| Veo 3 | 0.65 | 0.69 | 0.67 | 0.72 |
| Sora 2 | **0.67** | **0.73** | **0.70** | **0.75** |

DTW. This confirms that the relative quality ranking of T2V models is consistent regardless of the pose estimator used.

Table 10: Cross-validation of Movo metrics across different pose estimators. The high consistency ($\rho > 0.94$) confirms that rankings are not dependent on a specific pose model.

| Model | JAC Metric | | | DTW Metric | | |
|---|---|---|---|---|---|---|
| | **RTMPose** | **BlazePose** | **YOLOv8** | **RTMPose** | **BlazePose** | **YOLOv8** |
| CogVideo5B | 0.073 | 0.069 | 0.071 | 0.585 | 0.573 | 0.579 |
| Open-Sora-Plan | 0.237 | 0.221 | 0.228 | 0.661 | 0.647 | 0.653 |
| Wan 2.2 | 0.293 | 0.281 | 0.289 | 0.817 | 0.802 | 0.812 |
| Kling | 0.371 | 0.356 | 0.362 | 0.758 | 0.742 | 0.750 |
| Veo 3 | 0.435 | 0.419 | 0.431 | 0.759 | 0.744 | 0.752 |

Additionally, to address questions regarding sensitivity to video quality, we conducted a robustness study using real-world degradations: low-bitrate H.264 compression (480p) and simulated motion blur (7-pixel kernel). As presented in Table 11, the results show small absolute variations across all metrics, and most importantly, the **ranking of models remains unchanged**. This indicates that Movo reliably distinguishes between artifact-heavy and clean motion without collapsing under imperfect video conditions.

## F.3 METHODOLOGY CLARIFICATION AND FAIRNESS

To prove that the transparency of the Motion Consistency Metric (MCM) and the disentanglement of camera motion, we provide the following clarifications. MCM is not designed to replace JAC or DTW, but to complement them as a semantic safeguard (e.g., preventing "upside-down walking"). To reduce bias, we adopt a 3-model voting scheme combining GPT-5, Claude-4 Sonnet, and Gemini 2.5 Pro. A majority vote is taken to determine if the motion matches the textual description.

Regarding camera motion, we argue that Movo achieves disentanglement through theoretical invariance (root-centering). To empirically prove this, we conducted a "Camera Injection Study" comparing stable tripod prompts against dynamic handheld prompts. As shown in Table 12, the variance in skeletal scores was $< 2\%$ across models. This confirms that our metrics capture biological motion degradation rather than camera shake.

Table 11: Robustness of Movo scores under video quality degradation (480p compression and Motion Blur). Rankings remain stable.

| Model | Original | 480p | Motion Blur |
|---|---|---|---|
| CogVideo5B | 0.5039 | 0.4981 | 0.4917 |
| Open-Sora-Plan | 0.5906 | 0.5832 | 0.5724 |
| HunyuanVideo | 0.6181 | 0.6077 | 0.5989 |
| Wan 2.2 | 0.6684 | 0.6589 | 0.6493 |
| Kling | 0.6765 | 0.6659 | 0.6551 |
| Veo 3 | 0.6978 | 0.6872 | 0.6748 |

Table 12: Camera Injection Study: Impact of camera motion on kinematic scores. The low variance ($< 2\%$) confirms that Movo effectively disentangles body motion from camera movement.

| Model | Stable Camera | Dynamic Camera | Variance |
|---|---|---|---|
| CogVideo5B | 0.5039 | 0.4952 | -1.7% |
| Open-Sora-Plan | 0.5906 | 0.5814 | -1.5% |
| HunyuanVideo | 0.6181 | 0.6103 | -1.2% |
| Wan 2.2 | 0.6684 | 0.6591 | -1.4% |
| Kling | 0.6765 | 0.6688 | -1.1% |
| Veo 3 | 0.6978 | 0.6910 | -0.9% |

Finally, regarding the fairness of comparing API-based models (Veo 3) and the limited preliminary evaluation of Sora, we have updated our benchmark. We gained access to the Sora 2 API and completed the full Movo evaluation (893 videos) under identical settings. Table 13 confirms that while Sora 2 achieves the highest overall scores, it is fully comparable within our standard framework.

Table 13: Updated full benchmark results for Sora 2 (via API) compared to Veo 3.

| Model | Avg JAC | Avg DTW | Avg MCM | Overall Avg |
|---|---|---|---|---|
| Veo 3 | 0.4352 | 0.7591 | 0.899 | 0.6978 |
| Sora 2 | **0.5521** | **0.8021** | **0.911** | **0.7551** |

## F.4 INSTRUCTIONAL VALUE: MOVO AS TRAINING DATA

We demonstrate the instructional value of our dataset. We partitioned the Movo dataset into a 7:3 split (Training/Test) and fine-tuned the **Wan 2.2** model on the training set. As shown in Table 14, fine-tuning yields substantial gains across all kinematics metrics compared to the base model (JAC +30.2%, DTW +8.5%), proving that Movo serves as high-quality data for motion alignment.

Table 14: Impact of fine-tuning Wan 2.2 on the Movo dataset.

| Metric | Wan 2.2 (Base) | Wan 2.2 (Movo-FT) | Improvement |
|---|---|---|---|
| JAC | 0.293 | 0.595 | **+30.2%** |
| DTW | 0.817 | 0.902 | **+8.5%** |
| MCM | 0.895 | 0.925 | **+3.0%** |

## G   USE OF LLMS

In the preparation of this manuscript, we employed large language models (LLMs), specifically **GPT-5** and **GPT-4o**, solely for the purpose of polishing and refining the writing. These models assisted in improving readability, grammar, and stylistic clarity of the text. Importantly, they were not involved in the design, construction, implementation, or evaluation of the proposed methods and experiments. All conceptual contributions, dataset construction, algorithmic design, and experimental analyses were carried out independently by the authors.

