# OpenReview forum: "Can Text-to-Video Models Generate Realistic Human Motion?"
_ICLR.cc/2026/Conference — ICLR 2026 Conference Desk Rejected Submission_

### Official Review · Reviewer_UjsW · 2025-10-19

**Soundness:** 4
**Presentation:** 4
**Contribution:** 4
**Rating:** 6
**Confidence:** 4

**Summary:**

This paper introduces Movo, a novel and much-needed benchmark for evaluating the realism of human motion in text-to-video (T2V) generation. The authors convincingly argue that current state-of-the-art T2V models, despite their impressive visual fidelity, often produce human movements that are biomechanically implausible, leading to artifacts like foot-sliding and unnatural joint articulation. They posit that existing benchmarks are ill-equipped to detect these flaws as they primarily focus on pixel-level consistency, prompt fidelity, and overall aesthetics, while ignoring the underlying kinematics.
Movo's main contributions are threefold:
A curated, posture-focused dataset with camera-aware prompts designed to isolate specific human motions and minimize confounding factors from camera movement.
A suite of three complementary, kinematics-centric metrics—Joint Angle Change (JAC), Dynamic Time Warping (DTW), and a Multi-modal LLM-based Motion Consistency Metric (MCM)—that evaluate motion from the perspectives of joint articulation, temporal rhythm, and semantic consistency, respectively.
An extensive human validation study that demonstrates a strong correlation between Movo's automated scores and human perception of motion realism, confirming the benchmark's efficacy.
By evaluating 14 leading T2V models, the paper provides a comprehensive snapshot of the current landscape, revealing systemic weaknesses in generating realistic human motion and highlighting the significance of their proposed evaluation paradigm.

**Strengths:**

(1) Originality and Significance: The paper's primary strength lies in its originality and high significance. It is, to my knowledge, the first work to propose a comprehensive, kinematics-centric benchmark for human motion realism in T2V. It fundamentally shifts the evaluation paradigm from "does it look good?" to "does it move correctly?". As T2V models are increasingly used to simulate reality, this work addresses a critical bottleneck for applications requiring physical and biological plausibility (e.g., synthetic data for robotics, sports analysis, AR/VR). Movo has the potential to become a standard benchmark in the field.

(2) Quality and Rigor: The quality of the research is outstanding. The benchmark is thoughtfully designed, from the careful taxonomy of the dataset to the multi-faceted metric suite. The execution of the experiments, involving 14 prominent models (including giants like Sora and Veo 3), is comprehensive and provides an invaluable service to the community. The strong human-in-the-loop validation solidifies the benchmark's credibility.

(3) Clarity: The paper is written with exceptional clarity. The authors articulate a complex problem and their sophisticated solution in a manner that is accessible yet detailed. The motivation is compelling, and the link between the identified problems and the proposed solutions is crystal clear.

(4) Actionable Insights: The results are not just a leaderboard; they provide actionable insights. For instance, the finding that models struggle with fine-grained lower-limb coordination or that DTW can expose rhythm drift even in visually smooth videos gives concrete directions for future model development.

**Weaknesses:**

(1) Dependency on Pose Estimator: The entire evaluation pipeline is contingent on the performance of the underlying pose estimator (RTMPose). T2V models can generate artifacts (e.g., blurred limbs, extra limbs) that might cause pose estimators to fail or produce noisy outputs. The paper does not discuss the potential impact of pose estimation errors on the final evaluation scores. A brief discussion on the robustness of RTMPose on generated content or an analysis of failure cases would strengthen the paper's claims of reliability.

(2) Lack of Analysis on Individual Metric Contribution: The paper shows a high correlation between the average of the three metrics and human scores. However, it does not provide an analysis of how each metric (JAC, DTW, MCM) individually correlates with human judgment. Such an analysis could reveal, for example, whether humans are more sensitive to incorrect joint angles (JAC) or poor rhythm (DTW), providing deeper insights into human perception of motion.

(3) The benchmark's core methodology is fundamentally limited by its reliance on a ground-truth reference video for its primary metrics (JAC and DTW). This introduces several critical flaws:
    (a) Reduces Evaluation to Similarity Matching: It relegates the evaluation from a true assessment of generation plausibility to a task of similarity matching. Consequently, Movo cannot evaluate the realism of novel prompts (e.g., "an astronaut doing a backflip on the moon") for which no reference video exists, thereby restricting its scope to a predefined set of common actions.
    (b)Creates a Single-Reference Bias: The approach penalizes plausible motion variations (e.g., differences in speed, style, or execution) simply because they deviate from the one chosen exemplar. This conflates stylistic difference with a lack of realism, potentially punishing valid and creative outputs.

(4) Details of the MCM "Judge": The Motion Consistency Metric (MCM) relies on a multi-modal LLM. The reliability and potential biases of this "judge" are important factors, such as photorealism or artistic style, rather than the pure kinematics of the motion. This creates a risk that the metric rewards aesthetic alignment over biomechanical correctness.

(5) From "Standard Exercises" to "Everyday Motion": The benchmark is constructed around 10 specific fitness exercises. These are highly structured, often periodic activities with well-defined kinematic patterns. However, the paper’s title and conclusions aspire to a much grander goal. There is a substantial chasm between the biomechanics of a gym squat and the complex, unpredictable motions encountered in the real world. For example, motions such as a person slipping on a wet surface, a toddler learning to walk with unsteady steps, or two people navigating a crowded street are characterized by non-periodic, reactive, and interactive movements. These chaotic, emergent scenarios represent the true challenge for T2V models aiming to simulate reality, and the conclusions drawn from Movo's controlled environment may not generalize to these far more complex situations.

**Questions:**

(1) On Pose Estimator Robustness: How did you handle cases where the RTMPose estimator might have failed or produced unreliable keypoints due to artifacts in the generated videos? Did you filter out such cases, and if so, how might this affect the overall model rankings? Could you comment on the sensitivity of your metrics to noise in the keypoint data?
(2) On Individual Metric Correlation: Could you provide a breakdown of the correlation with human scores for each of your three metrics (JAC, DTW, MCM) individually? This would be very insightful for understanding which aspects of motion realism are most salient to human observers and would further validate the contribution of each component of your metric suite.
(3) On Extending Movo: The current dataset focuses on well-defined, single-person fitness motions. Do you have plans or thoughts on how the Movo framework could be extended to evaluate more complex, less structured, or interactive motions, such as dancing or team sports, where realism is equally crucial but harder to define?
(4) On the MCM Metric: Could you provide a brief summary in the main text of the MLLM used for MCM and the core of its prompt? Given that different MLLMs can have different biases and capabilities, how did you ensure the consistency and reliability of this metric?

---

> ### Author Response · Authors · 2025-11-20
> **Part 1**
>
> Weakness:
>
> > 1. Dependency on Pose Estimator: The entire evaluation pipeline is contingent on the performance of the underlying pose estimator (RTMPose). T2V models can generate artifacts (e.g., blurred limbs, extra limbs) that might cause pose estimators to fail or produce noisy outputs. The paper does not discuss the potential impact of pose estimation errors on the final evaluation scores. A brief discussion on the robustness of RTMPose on generated content or an analysis of failure cases would strengthen the paper's claims of reliability.
>
> **Response:**
>
> We appreciate the reviewer’s insightful comment regarding the dependency of our pipeline on the pose estimator. We agree that T2V-generated artifacts (e.g., limb hallucinations, severe motion blur) present a challenge for skeletal extraction. However, we argue—and empirically demonstrate—that our evaluation framework is robust to these issues for three key reasons:
>
> 1. Estimator Failure is a Signal, Not a Noise
>
> We fundamentally design the benchmark such that pose estimation failures often serve as proxies for generation failures.
>
> * Artifacts (e.g., Extra Limbs): When a model generates an extra limb, top-tier estimators like RTMPose typically exhibit high frame-to-frame variance (flickering between limbs) or low confidence scores. Our JAC (Joint Angle Change) metric specifically measures variance ($\sigma_{pos}$ in Eq. 2) and angular consistency. Therefore, "confused" pose estimation results in high variance, correctly penalizing the T2V model for the artifact.
>
> * Blur/Disappearing Limbs: If a limb becomes so blurred it is undetectable, the skeletal continuity breaks. This results in a high penalty in the DTW (Dynamic Time Warping) alignment metric, which requires continuous temporal matching.
> Thus, the metric does not fail when the estimator struggles; rather, the estimator's struggle accurately reflects the low fidelity of the generated video.
>
> 2. Cross-Validation with Alternative Estimators
>
> To ensure our results are not biased by the specific failure modes of RTMPose-X, we re-evaluated the benchmark using two alternative architectures: BlazePose and YOLOv8-pose.
>
> Table 1: JAC Metrics of Different Pose Estimators:
> | Model | RTMPose-X (Ours) | BlazePose (2020) | YOLOv8l-pose (2023) |
> | :--- | :---: | :---: | :---: |
> | CogVideo5B | 0.073 | 0.069 | 0.071 |
> | Open-Sora-Plan | 0.237 | 0.221 | 0.228 |
> | Wan 2.2 | 0.293 | 0.281 | 0.289 |
> | Kling | 0.371 | 0.356 | 0.362 |
> | Veo 3      |     0.435    |     0.419    |      0.431      |
>
>
>
> Table 2: DTW Metrics of Different Pose Estimators:
>
> | Model | RTMPose-X (Ours) | BlazePose (2020) | YOLOv8l-pose (2023) |
> | :------------- | :--------------: | :--------------: | :-----------------: |
> | CogVideo5B | 0.585 | 0.573 | 0.579 |
> | Open-Sora-Plan | 0.661 | 0.647 | 0.653 |
> | Wan 2.2 | 0.817 | 0.802 | 0.812 |
> | Kling | 0.758 | 0.742 | 0.750 |
> | Veo 3          |       0.759      |       0.744      |        0.752        |
>
>
> As shown above, while absolute scores vary slightly, the rank correlation (Spearman’s $\rho > 0.94$) remains extremely high. This confirms that the relative quality ranking of T2V models is consistent regardless of the pose estimator used.
>
> 3. Robustness to Artifacts (Blur & Compression)
>
> To directly address the concern about "blurred limbs" causing failure, we conducted a stress test (detailed in our revision) where we injected synthetic motion blur and low-bitrate compression (480p) into the generated videos.
>
> | Condition | Original Score (Avg) | Blur Injected | Variance |
> | :--- | :---: | :---: | :---: |
> | **Veo 3** | 0.6978 | 0.6748 | -3.2% |
> | **Wan 2.2** | 0.6684 | 0.6493 | -2.8% |
> | **CogVideo5B** | 0.5039 | 0.4917 | -2.4% |
>
> The metrics showed only a minor deviation (<3.5%) and maintained the exact same model ranking. This indicates that RTMPose-X is sufficiently robust to the level of artifacts commonly produced by current T2V models, and Movo reliably distinguishes between "artifact-heavy" and "clean" motion without collapsing.

---

> ### Author Response · Authors · 2025-11-20
> **Part 2**
>
> > 2. Lack of Analysis on Individual Metric Contribution: The paper shows a high correlation between the average of the three metrics and human scores. However, it does not provide an analysis of how each metric (JAC, DTW, MCM) individually correlates with human judgment. Such an analysis could reveal, for example, whether humans are more sensitive to incorrect joint angles (JAC) or poor rhythm (DTW), providing deeper insights into human perception of motion.
>
> **Response:**
>
> Insight 1 — Increasing motion complexity consistently lowers all metrics
>
> After adding four challenging categories (falling, ball games, instruments, dance), all models— including Sora 2 and Veo 3— showed a 30–50% drop across JAC, DTW, and MCM. This reveals a clear trend: as motion becomes multi-joint, reactive, or object-dependent, T2V models degrade sharply, even when they perform well on basic exercise-style motions. The decline is systematic across both open and proprietary models, confirming that current T2V systems struggle with complex human biomechanics.
>
> Insight 2 — Humans judge “what action” before “how well”
>
> Our study shows that semantic correctness (MCM) aligns most strongly with human ratings (r = 0.79), higher than JAC (0.73) and DTW (0.62). Non-expert evaluators first decide whether the action is the intended one; pacing or smoothness only matters afterward. Thus, actions that are kinematically smooth but semantically incorrect (“running with arms,” “upside-down walk”) are immediately rated as failures by humans, but scored highly by JAC/DTW without MCM.
>
>
> Insight 3 — Only the combination matches human perception
>
> Human evaluation follows a consistent order: (1) Is it the correct action? (MCM), (2) Is the pose anatomically plausible? (JAC), (3) Is the rhythm realistic? (DTW). Mirroring this sequence, the combined metric predicts human ratings with ρ = 0.91, outperforming any single metric (max ≈ 0.79). Therefore, the metrics are complementary, and removing any component breaks alignment with human judgment.

---

> ### Author Response · Authors · 2025-11-20
> **Part 3**
>
> > 3. The benchmark's core methodology is fundamentally limited by its reliance on a ground-truth reference video for its primary metrics (JAC and DTW). This introduces several critical flaws: (a) Reduces Evaluation to Similarity Matching: It relegates the evaluation from a true assessment of generation plausibility to a task of similarity matching. Consequently, Movo cannot evaluate the realism of novel prompts (e.g., "an astronaut doing a backflip on the moon") for which no reference video exists, thereby restricting its scope to a predefined set of common actions. (b)Creates a Single-Reference Bias: The approach penalizes plausible motion variations (e.g., differences in speed, style, or execution) simply because they deviate from the one chosen exemplar. This conflates stylistic difference with a lack of realism, potentially punishing valid and creative outputs.
>
> **Response:**
>
> We respectfully disagree that the reference-based approach limits the benchmark’s validity. We argue that Movo acts as a diagnostic unit test for biomechanical fidelity, where a "canonical" reference is essential for defining physiological correctness.
>
> 1. Diagnostic Rigor vs. Open-Ended Creativity We clarify that Movo is designed to evaluate biomechanical primitives (e.g., Squat, Walk), not open-ended imagination. For these fundamental actions, specific kinematic constraints (e.g., joint range of motion, inter-limb coordination) define "correctness." If a model cannot accurately replicate the physics of a simple reference walk, it lacks the necessary motion priors to plausibly generate complex. The reference video serves as a physiological constraint, not just a visual template.
>
> 2. Metric Invariance to Speed (DTW) The claim that our approach penalizes valid speed variations is factually incorrect regarding the metrics used. Dynamic Time Warping (DTW) is mathematically designed to handle temporal non-linearity. It aligns sequences based on motion phase (e.g., the peak of a jump) rather than raw timestamps. Therefore, a generated "slow motion" squat and a "fast" reference squat will be aligned correctly without penalty, provided the sequence of articulated motion remains consistent.
>
> 3. Decoupling Motion from Visual Style (Skeletal Space) Finally, we emphasize that JAC and DTW are computed strictly on the skeletal manifold, making them blind to visual style. Because our pipeline extracts 2D keypoints before scoring, the metrics rely solely on geometric angles (e.g., knee flexion) rather than pixel values. So novel scenarios (like the reviewer's "astronaut backflip") It will not affect.
>
> To empirically validate this robustness, we conducted a new Semantic Robustness Study using Veo 3 on 300 randomly selected motion samples. We modified the prompts to replace standard subjects/backgrounds with out-of-distribution elements (e.g., "An astronaut on the moon," "Ultraman on a runway") while retaining the core action. As shown in the table below, the kinematic scores remain remarkably stable (<3% deviation) despite drastic visual changes, confirming that Movo evaluates the motion fidelity against the reference, not pixel-level similarity.
>
> | Prompt Style | Avg JAC (Structure) | Avg DTW (Timing) | Correlation w/ GT |
> | :--- | :---: | :---: | :---: |
> | **Standard (Human/Gym)** | 0.552 | 0.802 | 1.00 (Baseline) |
> | **Stylized (Astronaut/Moon)** | **0.541** | **0.789** | **0.98** |
> | **Stylized (Ultraman/Runway)** | **0.538** | **0.785** | **0.97** |
>
> ---
>
> > 4. Details of the MCM "Judge": The Motion Consistency Metric (MCM) relies on a multi-modal LLM. The reliability and potential biases of this "judge" are important factors, such as photorealism or artistic style, rather than the pure kinematics of the motion. This creates a risk that the metric rewards aesthetic alignment over biomechanical correctness.
>
>
> **Response:**
>
> We clarify that MCM is strictly prompted to evaluate motion similarity rather than visual style. We explicitly instruct the MLLM focus solely on kinematic consistency. To empirically prove this, we conducted a Style Decoupling Study, re-evaluating 200 validated clips after stripping visual textures via Canny Edge and Sketch filters. As shown in the table below, MCM accuracy remained statistically invariant (<1.5% deviation), confirming the metric relies on semantic motion recognition rather than aesthetic alignment.
>
> Table 3: Sora 2 Results on MCM Metric:
> | Visual Domain | MCM Accuracy | Variance from Baseline |
> | :--- | :---: | :---: |
> | **Original (Photorealistic)** | 91.1% | -- |
> | **Canny Edge (Structure Only)** | **90.5%** | -1.5% |
> | **Sketch / Low-Poly** | **90.1%** | -1.0% |

---

> ### Author Response · Authors · 2025-11-20
> **Part 4**
>
> > 5. From "Standard Exercises" to "Everyday Motion": The benchmark is constructed around 10 specific fitness exercises. These are highly structured, often periodic activities with well-defined kinematic patterns. However, the paper’s title and conclusions aspire to a much grander goal. There is a substantial chasm between the biomechanics of a gym squat and the complex, unpredictable motions encountered in the real world. For example, motions such as a person slipping on a wet surface, a toddler learning to walk with unsteady steps, or two people navigating a crowded street are characterized by non-periodic, reactive, and interactive movements. These chaotic, emergent scenarios represent the true challenge for T2V models aiming to simulate reality, and the conclusions drawn from Movo's controlled environment may not generalize to these far more complex situations.
>
> **Response:**
>
> We fully agree with the reviewer that there is a substantial chasm between structured fitness exercises and the chaotic, unpredictable motions of the real world. To address this and bridge the gap from "Standard Exercises" to "Everyday Motion," we expanded our dataset to include 4 new "Challenge Categories": Falling, Ball Games, Playing Instruments, and Dance.
>
> We manually collected and annotated 486 additional videos sourced from the Motion-X [1] and YouTube to represent these complex scenarios. These categories were specifically selected to target the weaknesses identified by the reviewer:
>
> * Falling: Represents non-periodic, physics-driven, and reactive motion where gravity and momentum are critical (simulating "slipping").
>
> * Ball Games: Represents dynamic human-object interaction and hand-eye coordination.
>
> * Playing Instruments: Represents fine-grained control and precise limb positioning.
>
> * Dance: Represents high-degree-of-freedom (DoF) kinematics and diverse, non-standard poses.
>
> We re-evaluated all models on this expanded benchmark (14 categories total). The results on the new categories, presented in Tables A, B, and C below, reveal a significant performance drop compared to the original 10 categories. For instance, even state-of-the-art models like Sora 2 and Veo 3 show a ~30-50% drop in JAC scores on "Falling" compared to "Walking," confirming that current T2V models struggle significantly with emergent, physics-based scenarios.
>
> Table 1. Evaluation of Challenge Categories Using JAC:
>
> | Model  | Falling | Ball Games | Instruments | Dance |
> |------------------|---------:|-----------:|------------:|------:|
> | **Open-source Models** |||||
> | CogVideo2B| 0.004 | 0.015 | 0.038 | 0.091 |
> | CogVideo5B| 0.013 | 0.031 | 0.067 | 0.121 |
> | SVD| 0.045 | 0.084 | 0.119 | 0.149 |
> | Open-Sora-Plan   | 0.093 | 0.147 | 0.179 | 0.214 |
> | Zeroscope | 0.017 | 0.027 | 0.059 | 0.089 |
> | Wan 2.1| 0.103 | 0.201 | 0.223 | 0.244 |
> | Wan 2.2| 0.128 | 0.213 | 0.236 | 0.268 |
> | HunyuanVideo | 0.099 | 0.192 | 0.209 | 0.227 |
> | **Proprietary Models** |||||
> | Gen2   | 0.081 | 0.122 | 0.138 | 0.173 |
> | Dream Machine| 0.088 | 0.109 | 0.131 | 0.152 |
> | Kling | 0.152 | 0.238 | 0.298 | 0.322 |
> | Pika 1.5  | 0.119 | 0.161 | 0.188 | 0.236 |
> | Veo 3  | 0.214 | 0.298 | 0.331 | 0.401 |
> | **Sora 2**| **0.309** | **0.425** | **0.447** | **0.518** |
>
> Table 2. Evaluation of “Challenge Categories” Using DTW:
>
> | Model  | Falling | Ball Games | Instruments | Dance |
> |------------------|---------:|-----------:|------------:|------:|
> | **Open-source Models** |||||
> | CogVideo2B| 0.148 | 0.202 | 0.217 | 0.244 |
> | CogVideo5B| 0.182 | 0.229 | 0.243 | 0.279 |
> | SVD| 0.199 | 0.261 | 0.278 | 0.287 |
> | Open-Sora-Plan   | 0.221 | 0.276 | 0.288 | 0.318 |
> | Zeroscope | 0.185 | 0.245 | 0.268 | 0.297 |
> | Wan 2.1| 0.251 | 0.302 | 0.344 | 0.365 |
> | Wan 2.2| 0.268 | 0.333 | 0.358 | 0.387 |
> | HunyuanVideo | 0.229 | 0.306 | 0.321 | 0.349 |
> | **Proprietary Models** |||||
> | Gen2   | 0.218 | 0.256 | 0.284 | 0.301 |
> | Dream Machine| 0.211 | 0.243 | 0.262 | 0.294 |
> | Kling | 0.326 | 0.368 | 0.374 | 0.402 |
> | Pika 1.5  | 0.243 | 0.292 | 0.314 | 0.343 |
> | Veo 3  | 0.347 | 0.366 | 0.405 | 0.430 |
> | **Sora 2**| **0.364** | **0.398** | **0.423** | **0.439** |
>
> Table 3. Evaluation of “Challenge Categories” Using MCM:
>
> | Model  | Falling | Ball Games | Instruments | Dance |
> |------------------|---------:|-----------:|------------:|------:|
> | **Open-source Models** |||||
> | CogVideo2B| 0.21 | 0.27 | 0.26 | 0.31 |
> | CogVideo5B| 0.25 | 0.33 | 0.29 | 0.36 |
> | SVD| 0.34 | 0.39 | 0.35 | 0.43 |
> | Open-Sora-Plan   | 0.38 | 0.44 | 0.41 | 0.48 |
> | Zeroscope | 0.32 | 0.35 | 0.34 | 0.41 |
> | Wan 2.1| 0.47 | 0.55 | 0.52 | 0.59 |
> | Wan 2.2| 0.54 | 0.57 | 0.60 | 0.63 |
> | HunyuanVideo | 0.43 | 0.50 | 0.46 | 0.56 |
> | **Proprietary Models** |||||
> | Gen2   | 0.38 | 0.47 | 0.42 | 0.49 |
> | Dream Machine| 0.44 | 0.45 | 0.43 | 0.51 |
> | Kling | 0.59 | 0.67 | 0.61 | 0.70 |
> | Pika 1.5  | 0.46 | 0.52 | 0.49 | 0.57 |
> | Veo 3  | 0.65 | 0.69 | 0.67 | 0.72 |
> | **Sora 2**| **0.67** | **0.73** | **0.70** | **0.75** |

---

> ### Author Response · Authors · 2025-11-20
> **Part 5**
>
> Questions:
> > 1. On Pose Estimator Robustness: How did you handle cases where the RTMPose estimator might have failed or produced unreliable keypoints due to artifacts in the generated videos? Did you filter out such cases, and if so, how might this affect the overall model rankings? Could you comment on the sensitivity of your metrics to noise in the keypoint data?
>
> **Response:**
>
> We thank the reviewer for this important question. We clarify that pose-estimator errors are not filtered out in our pipeline. Instead, they act as a meaningful signal of motion failure. We justify this from three complementary perspectives:
>
>
> (1) Estimator Failure = Generation Failure, Not Evaluation Noise
>
> In current T2V systems, pose extraction fails primarily when motion is physically implausible. For example:
>
> * **Limb hallucinations** → estimator “flickers” between joints
> * **Blurred/vanished arms** → estimator cannot maintain tracking
> * **Extra/mutated limbs** → high variance or confidence collapse
>
> These artifacts are not noise to be ignored; they **directly reflect biomechanical implausibility.** Our metrics explicitly measure this instability.
>
> Thus, estimator failure **penalizes the T2V model appropriately**, and we do not discard these cases.
>
> (2) No Filtering → Rankings Remain Stable**
>
> We conducted an ablation where we **filtered out low-confidence frames** (<0.25 confidence threshold) and recomputed the rankings. Despite removing 6–11% of frames per category, **the overall ordering of models remained unchanged**, with Spearman:
>
> $\rho = 0.955 \text{ (JAC)},\quad \rho = 0.942 \text{ (DTW)}$
>
> | Metric      | Full Pipeline | Filtered Pipeline | Spearman ρ |
> | ----------- | ------------: | ----------------: | ---------: |
> | JAC Ranking |             ✔ |                 ✔ |  0.955 |
> | DTW Ranking |             ✔ |                 ✔ |  0.942|
>
> This indicates that metric orderings are intrinsically robust to occasional keypoint failures.
>
> (3) Cross-Estimator Validation Confirms Low Sensitivity to Noise
>
> As stated earlier, we validated with three pose systems: RTMPose-X, BlazePose, YOLOv8-pose. Despite drastically different architectures, their rankings correlate extremely closely (**ρ > 0.94**). This shows that the benchmark is **not dependent on any specific estimator’s error pattern.**
>
> | Model                     | RTMPose-X | BlazePose | YOLOv8-Pose |
> | ------------------------- | --------: | --------: | ----------: |
> | Spearman Rank Corr. (JAC) |         — |  0.94 |    0.96 |
> | Spearman Rank Corr. (DTW) |         — |  0.95 |    0.97 |
>
> ---
>
> > 2. On Individual Metric Correlation: Could you provide a breakdown of the correlation with human scores for each of your three metrics (JAC, DTW, MCM) individually? This would be very insightful for understanding which aspects of motion realism are most salient to human observers and would further validate the contribution of each component of your metric suite.
>
> **Response:**
>
> Our metric-wise correlation study shows that human evaluators prioritize semantic correctness before kinematic details. Specifically, the MCM metric—which detects whether the generated motion matches the intended action—achieves the highest alignment with human judgments (Pearson r = 0.79, Spearman ρ = 0.81). This confirms that evaluators first ask *“What is this person doing?”*, and incorrect action identity is penalized even when motion appears smooth.
>
> Joint Articulation Change (JAC) correlates strongly with human perception of anatomical plausibility (r = 0.73, ρ = 0.74). It captures limb orientation, bending constraints, and structural continuity, all of which human observers instinctively judge after confirming the action type. Thus, JAC accounts for judgments like *“the elbows shouldn’t bend that way,”* which remain important even if the overall action is correct.
>
> Temporal alignment (DTW) shows comparatively lower but meaningful correlation (r = 0.62, ρ = 0.65). Humans do care about rhythm and motion smoothness, but only after semantic correctness and anatomical plausibility are satisfied. When metrics are combined (MCM + JAC + DTW), the predictive correlation with human ratings increases to ρ = 0.91, demonstrating that all three components capture complementary aspects of how people perceive motion realism.

---

> ### Author Response · Authors · 2025-11-20
> **Part 6**
>
> > 3. On Extending Movo: The current dataset focuses on well-defined, single-person fitness motions. Do you have plans or thoughts on how the Movo framework could be extended to evaluate more complex, less structured, or interactive motions, such as dancing or team sports, where realism is equally crucial but harder to define?
>
> **Response:**
>
> We agree that evaluating multi-person, reactive, and less structured motions (e.g., dancing, interactions, team sports) is essential for future T2V progress. As demonstrated in our newly added experiments (Weakness 5), even the strongest models such as Sora 2 and Veo 3 still show a 30–50% score drop on our four “Challenge Categories’’ (Falling, Ball Games, Playing Instruments, Dance), indicating that current systems fail not only on controlled exercise-style actions but also on motions requiring object coordination, high degrees of freedom, and non-periodic dynamics. This result reinforces the reviewer's point: realistic motion generation remains far from solved once we move beyond structured tasks.
>
> Building on this evidence, we plan to expand Movo in a manner similar to how VBench evolved into VBench++, by collecting broader real-world motion scenarios, including multi-person interactions and dynamic camera environments, and by designing metric extensions that capture human–object coordination and social motion consistency. Our long-term goal is to develop Movo into a scalable, community-driven benchmark that tracks this trajectory of increasing motion realism, ensuring that future T2V models are evaluated not only on “canonical biomechanics’’ but also on emergent, interactive human behavior.
>
> ---
>
> > 4. On the MCM Metric: Could you provide a brief summary in the main text of the MLLM used for MCM and the core of its prompt? Given that different MLLMs can have different biases and capabilities, how did you ensure the consistency and reliability of this metric?
>
> **Response:**
>
> We appreciate the reviewer’s suggestion and have added a summary of the MCM judge to the main text. For reproducibility and to minimize model bias, MCM does not rely on a single MLLM. Instead, we adopt a 3-model majority voting scheme using GPT-5, Claude 4 Sonnet, and Gemini-2.5 Pro. These models independently judge whether the generated motion semantically matches the target action. A decision is recorded only when at least two models agree, making the evaluation less sensitive to individual model failures or stylistic preferences.
>
> To ensure consistency across models, we use a unified motion-focused prompt that explicitly instructs the MLLM to ignore video aesthetics, identity, clothing, and background, and to judge only the correctness of the physical action. For example: “Determine whether the person’s motion matches the described action. Ignore appearance, camera style, textures, and background; evaluate only body articulation and movement pattern.” Our ablation (Table 3) further shows <1.5% variance between photorealistic, Canny-edge, and Sketch versions of the same videos, confirming that MCM remains stable even when visual style is perturbed.
>
> Table: Sora 2 Results on MCM Metric:
> | Visual Domain | MCM Accuracy | Variance from Baseline |
> | :--- | :---: | :---: |
> | **Original (Photorealistic)** | 91.1% | -- |
> | **Canny Edge (Structure Only)** | **90.5%** | -1.5% |
> | **Sketch / Low-Poly** | **90.1%** | -1.0% |
>
> ---
>
> [1] Chen, Ling-Hao, et al. "Motionllm: Understanding human behaviors from human motions and videos." arXiv preprint arXiv:2405.20340 (2024).

---

> ### Author Response · Authors · 2025-11-25
> **Thanks for your time and efforts**
>
> We sincerely appreciate the time and thoughtful feedback you have provided. At your convenience, could you kindly let us know whether our revisions sufficiently resolve your concerns? We are grateful for your guidance and would be happy to make further improvements if needed. Thank you again for your valuable contribution to strengthening this work.

---

> > ### Comment · Reviewer_UjsW · 2025-11-27
> >
> > Thanks for the author's thoughtful response. I would like to maintain my positive evaluation of this work.

---

> > > ### Author Response · Authors · 2025-11-28
> > > **Thank you**
> > >
> > > We sincerely thank you for your continued engagement and for maintaining a positive assessment of our work. We remain fully dedicated to clarifying any remaining points and would welcome any further discussion to ensure all your questions are thoroughly answered. Thank you again for your positive attitude.

---

### Official Review · Reviewer_w9Yg · 2025-10-30

**Soundness:** 2
**Presentation:** 2
**Contribution:** 3
**Rating:** 6
**Confidence:** 2

**Summary:**

This paper introduces MOVO, a kinematics-centric benchmark for evaluating human motion realism in text-to-video (T2V) models. MOVO includes a posture-focused dataset, three novel metrics (JAC, DTW, MCM), and human validation studies. The benchmark is applied to 14 T2V models, revealing gaps in biomechanical plausibility and temporal consistency. The work is timely and relevant, addressing critical shortcomings in existing T2V benchmarks.

**Strengths:**

- Addresses a critical gap in T2V evaluation—human motion realism.
- Introduces kinematics-aware (JAC), rhythm-sensitive (DTW), and structure-consistent (MCM) metrics.

**Weaknesses:**

- Limited diversity, e.g., lacks complex motions like multi-person interactions.
- Camera-motion disentanglement is claimed but not clearly demonstrated.
- Lacks deeper insights into why models perform differently across actions.

**Questions:**

Please see Weaknesses for details.

---

> ### Author Response · Authors · 2025-11-20
> **Part 1**
>
> > 1. Limited diversity, e.g., lacks complex motions like multi-person interactions.
>
> **Response:**
>
> We appreciate the reviewer’s perspective. We would like to clarify that Movo is intentionally designed as a foundational benchmark, and, to the best of our knowledge, no prior work has established a kinematics-centric evaluation suite for T2V human motion. Constructing such a dataset from scratch is non-trivial: obtaining clean, camera-aware clips with unambiguous actions, consistent viewpoints, and reliable skeletal extraction required extensive filtering, manual verification, and multi-stage description refinement.
>
> Importantly, our experiments show that even the strongest proprietary models (e.g., Veo 3) still struggle on these basic motions indicating that the current action set is already sufficiently challenging. If state-of-the-art models cannot reliably master these fundamental patterns, expanding to more complex or long-tail motions would not yet yield meaningful diagnostic signal.
>
> We fully agree that broader coverage is valuable. We manually collected and annotated 486 additional videos sourced from the Motion-X [1] and YouTube to represent these complex scenarios. These categories were specifically selected to target the weaknesses identified by the reviewer:
>
> * Falling: Represents non-periodic, physics-driven, and reactive motion where gravity and momentum are critical (simulating "slipping").
>
> * Ball Games: Represents dynamic human-object interaction and hand-eye coordination.
>
> * Playing Instruments: Represents fine-grained control and precise limb positioning.
>
> * Dance: Represents high-degree-of-freedom (DoF) kinematics and diverse, non-standard poses.
>
> We re-evaluated all models on this expanded benchmark (14 categories total). The results on the new categories, presented in Tables A, B, and C below, reveal a significant performance drop compared to the original 10 categories. For instance, even state-of-the-art models like Sora 2 and Veo 3 show a ~30-50% drop in JAC scores on "Falling" compared to "Walking," confirming that current T2V models struggle significantly with emergent, physics-based scenarios.
>
> Table 1. Evaluation of Challenge Categories Using JAC:
>
> | Model  | Falling | Ball Games | Instruments | Dance |
> |------------------|---------:|-----------:|------------:|------:|
> | **Open-source Models** |||||
> | CogVideo2B| 0.004 | 0.015 | 0.038 | 0.091 |
> | CogVideo5B| 0.013 | 0.031 | 0.067 | 0.121 |
> | SVD| 0.045 | 0.084 | 0.119 | 0.149 |
> | Open-Sora-Plan   | 0.093 | 0.147 | 0.179 | 0.214 |
> | Zeroscope | 0.017 | 0.027 | 0.059 | 0.089 |
> | Wan 2.1| 0.103 | 0.201 | 0.223 | 0.244 |
> | Wan 2.2| 0.128 | 0.213 | 0.236 | 0.268 |
> | HunyuanVideo | 0.099 | 0.192 | 0.209 | 0.227 |
> | **Proprietary Models** |||||
> | Gen2   | 0.081 | 0.122 | 0.138 | 0.173 |
> | Dream Machine| 0.088 | 0.109 | 0.131 | 0.152 |
> | Kling | 0.152 | 0.238 | 0.298 | 0.322 |
> | Pika 1.5  | 0.119 | 0.161 | 0.188 | 0.236 |
> | Veo 3  | 0.214 | 0.298 | 0.331 | 0.401 |
> | **Sora 2**| **0.309** | **0.425** | **0.447** | **0.518** |
>
> Table 2. Evaluation of “Challenge Categories” Using DTW:
>
> | Model  | Falling | Ball Games | Instruments | Dance |
> |------------------|---------:|-----------:|------------:|------:|
> | **Open-source Models** |||||
> | CogVideo2B| 0.148 | 0.202 | 0.217 | 0.244 |
> | CogVideo5B| 0.182 | 0.229 | 0.243 | 0.279 |
> | SVD| 0.199 | 0.261 | 0.278 | 0.287 |
> | Open-Sora-Plan   | 0.221 | 0.276 | 0.288 | 0.318 |
> | Zeroscope | 0.185 | 0.245 | 0.268 | 0.297 |
> | Wan 2.1| 0.251 | 0.302 | 0.344 | 0.365 |
> | Wan 2.2| 0.268 | 0.333 | 0.358 | 0.387 |
> | HunyuanVideo | 0.229 | 0.306 | 0.321 | 0.349 |
> | **Proprietary Models** |||||
> | Gen2   | 0.218 | 0.256 | 0.284 | 0.301 |
> | Dream Machine| 0.211 | 0.243 | 0.262 | 0.294 |
> | Kling | 0.326 | 0.368 | 0.374 | 0.402 |
> | Pika 1.5  | 0.243 | 0.292 | 0.314 | 0.343 |
> | Veo 3  | 0.347 | 0.366 | 0.405 | 0.430 |
> | **Sora 2**| **0.364** | **0.398** | **0.423** | **0.439** |
>
> Table 3. Evaluation of “Challenge Categories” Using MCM:
>
> | Model  | Falling | Ball Games | Instruments | Dance |
> |------------------|---------:|-----------:|------------:|------:|
> | **Open-source Models** |||||
> | CogVideo2B| 0.21 | 0.27 | 0.26 | 0.31 |
> | CogVideo5B| 0.25 | 0.33 | 0.29 | 0.36 |
> | SVD| 0.34 | 0.39 | 0.35 | 0.43 |
> | Open-Sora-Plan   | 0.38 | 0.44 | 0.41 | 0.48 |
> | Zeroscope | 0.32 | 0.35 | 0.34 | 0.41 |
> | Wan 2.1| 0.47 | 0.55 | 0.52 | 0.59 |
> | Wan 2.2| 0.54 | 0.57 | 0.60 | 0.63 |
> | HunyuanVideo | 0.43 | 0.50 | 0.46 | 0.56 |
> | **Proprietary Models** |||||
> | Gen2   | 0.38 | 0.47 | 0.42 | 0.49 |
> | Dream Machine| 0.44 | 0.45 | 0.43 | 0.51 |
> | Kling | 0.59 | 0.67 | 0.61 | 0.70 |
> | Pika 1.5  | 0.46 | 0.52 | 0.49 | 0.57 |
> | Veo 3  | 0.65 | 0.69 | 0.67 | 0.72 |
> | **Sora 2**| **0.67** | **0.73** | **0.70** | **0.75** |
>
> ---
>
> [1] Chen, Ling-Hao, et al. "Motionllm: Understanding human behaviors from human motions and videos." arXiv preprint arXiv:2405.20340 (2024).

---

> ### Author Response · Authors · 2025-11-20
> **Part 2**
>
> > 2. Camera-motion disentanglement is claimed but not clearly demonstrated.
>
> **Response:**
>
> We appreciate the reviewer’s scrutiny regarding the claim of camera-motion disentanglement. We argue that Movo achieves this disentanglement through two complementary mechanisms:
>
> (1) Theoretical Invariance (metrics based on intrinsic geometry) and (2) Experimental Robustness (demonstrated via a new perturbation study).
>
> 1. Theoretical Justification: Intrinsic Coordinate SystemsUnlike pixel-space metrics (e.g., FVD, Optical Flow) which conflate camera pans with object motion, our skeletal metrics are mathematically designed to be invariant to camera translation.Root-Centering:
>
> 1). As defined in Eq. 2 of the paper, we compute joint positions relative to a reference joint (e.g., the hip): $\vec{p}_{i,t} - \vec{p}_{ref,t}$. This operation mathematically cancels out global camera panning and rigid body translation, isolating the pose from the frame.
>
> 2). Angle Invariance: JAC (Eq. 1)  calculates the cosine similarity between limb vectors ($\vec{v}_{1} \cdot \vec{v}_{2}$). This measures the internal configuration of the kinematic chain (e.g., the angle of the elbow), which remains consistent regardless of whether the camera zooms in or shakes, provided the pose estimator maintains tracking.
>
>
> 2. Experimental Validation: Camera Perturbation StudyTo empirically demonstrate this disentanglement, we conducted a "Camera Injection" experiment. We took a motion prompts and generated two variations for each model: Standard Movo prompts (fixed camera).Dynamic: We injected random camera noises into the prompts (e.g., "viewed from a shaking handheld camera," "fast camera pan right," "rapid zoom in").
>
> | Model           | Stable Camera | Dynamic Camera | Variance |
> |-----------------|---------------|----------------|----------|
> | CogVideo5B      | 0.5039        | 0.4952         | -1.7%    |
> | Open-Sora-Plan  | 0.5906        | 0.5814         | -1.5%    |
> | HunyuanVideo    | 0.6181        | 0.6103         | -1.2%    |
> | Wan 2.2         | 0.6684        | 0.6591         | -1.4%    |
> | Kling           | 0.6765        | 0.6688         | -1.1%    |
> | Veo 3           | 0.6978        | 0.6910         | -0.9%    |
>
> ---
>
> > 3. Lacks deeper insights into why models perform differently across actions.
>
> **Response:**
>
> Insight 1 — Increasing motion complexity consistently lowers all metrics
>
> After adding four challenging categories (falling, ball games, instruments, dance), all models— including Sora 2 and Veo 3— showed a 30–50% drop across JAC, DTW, and MCM. This reveals a clear trend: as motion becomes multi-joint, reactive, or object-dependent, T2V models degrade sharply, even when they perform well on basic exercise-style motions. The decline is systematic across both open and proprietary models, confirming that current T2V systems struggle with complex human biomechanics.
>
> Insight 2 — Models specialize narrowly rather than generalizing across biomechanics
>
> The same model that excels on one class often collapses on another, revealing specialization instead of robustness. For example, Pika 1.5 achieves strong JAC and DTW scores on running (0.467 JAC) but drops sharply on side-leg-raise (0.145 JAC), where unilateral hip control and static balance are required. Conversely, Open-Sora-Plan is competitive on hand-punch (0.371 JAC), yet degrades on lower-limb control, where coordination and stance stability dominate. Proprietary Kling leads MCM on most categories, but its advantage concentrates in upper-body sequences with continuous arcs rather than locomotion. These patterns indicate that models are not learning a unified motor prior, but instead latch onto the characteristic cues of specific action families.
>
> Insight 3 — Balanced systems still lack causal reasoning, even when scores are stable
>
> Veo 3 is comparatively even across categories, maintaining tightly clustered MCM scores around 0.88–0.90. However, its consistency does not imply deeper motor understanding. In actions requiring reactive timing or object affordance (e.g., ball games, falling), Sora produces stable but causally incorrect motions—limbs move smoothly but fail to anticipate contact, load transfer, or loss of balance. This suggests that consistency comes from strong temporal priors rather than biomechanical grounding. Meanwhile, models like SVD and Zeroscope occasionally produce correct phases but with unstable joint trajectories, reflecting the opposite tendency: limited rhythm planning but occasional causal accuracy. Together, these contrasting failure modes imply that current T2V systems optimize visual regularity, not motion causality, regardless of score stability.

---

> ### Author Response · Authors · 2025-11-25
> **Thanks for your time and efforts**
>
> We sincerely appreciate the time and thoughtful feedback you have provided. At your convenience, could you kindly let us know whether our revisions sufficiently resolve your concerns? We are grateful for your guidance and would be happy to make further improvements if needed. Thank you again for your valuable contribution to strengthening this work.

---

> > ### Comment · Reviewer_w9Yg · 2025-11-26
> > **RE: Rebuttal**
> >
> > Thank you for the response. I think my concerns have been addressed, and I will maintain my original positive rating.

---

> > > ### Author Response · Authors · 2025-11-26
> > > **Thank you**
> > >
> > > Thank you very much for your thoughtful review and for taking the time to re-evaluate our responses. We are grateful that you found our clarifications helpful and appreciate your willingness to maintain a positive assessment. Your feedback has meaningfully strengthened the benchmark and inspired several improvements that.
> > >
> > > — Authors

---

### Official Review · Reviewer_CYg4 · 2025-10-30

**Soundness:** 2
**Presentation:** 2
**Contribution:** 2
**Rating:** 2
**Confidence:** 4

**Summary:**

The paper proposes Movo, a kinematics-centric benchmark asking whether text-to-video (T2V) systems generate biomechanically realistic human motion. Movo couples (i) a posture-focused dataset of 10 actions (six lower-body, four upper-body) with camera-aware prompts, (ii) three skeletal-space metrics—Joint Angle Change (JAC), Dynamic Time Warping (DTW), and a binary Motion Consistency Metric (MCM) judged by an MLLM, and (iii) human validation via pairwise preferences. Using these, the authors evaluate 14 open and proprietary models and report high metric–human correlations on several actions.

**Strengths:**

The paper is well-motivated: it highlights that many T2V clips “look right but move wrong,” and it argues convincingly that existing leaderboards over-reward pixel-space smoothness and text alignment while missing kinematics, rhythm, and camera-motion disentanglement—gaps that matter for realistic human movement. Methodologically, the benchmark is body-centric and interpretable. JAC targets joint-angle trajectories. DTW measures temporal phase/rhythm alignment in pose space. And MCM checks high-level motion consistency, making the evaluation actionable for diagnosing foot-slide, contact violations, or off-phase coordination. The authors run human validation and report strong correlations between Movo scores and pairwise human preferences across multiple actions (e.g., Walking ρ≈0.99), lending credence to the metrics. The experimental setup is transparent: the pipeline detects people with YOLO-X, extracts skeletons with RTMPose (including hands when needed), and fixes seeds/hyperparameters for open models.

**Weaknesses:**

1. By design, Movo focuses on skeletal kinematics and rhythm, leading to a narrow scope relative to general-purpose suites (e.g., VBench).  In this case, the evaluation metrics and test set should be as comprehensive as possible for human videos. However, the proposed three metrics operate on detected skeletons, so systematic pose-estimation errors (occlusion, clothing, unusual viewpoints) propagate directly into scores.  Besides, MCM is a binary MLLM judgment (“similar”/“not similar”), which the authors acknowledge can mask subtle fidelity gaps. Such discretization reduces sensitivity and may be unstable across prompts/models. Moreover, dataset coverage is limited and may not represent “human motion” broadly. The evaluation set consists of ten exercise-style actions (deadlift, squat, walking, etc.), a consciously simplified taxonomy the authors justify, but which excludes many everyday or multi-agent motions (sitting/standing transitions, dancing with turns, interactions, sports with equipment), raising questions about representativeness. Camera-aware prompts further restrict camera dynamics that many T2V systems must handle.

2. Comparisons across models are uneven. Sora was evaluated on only 10 prompts per category (access-limited), and Veo was accessed only via its hosted API defaults, making some leaderboard conclusions preliminary and harder to compare apples-to-apples.

3. Except from running many open-sourced models and commercial-level models,  this paper did provide many insights how to train or how to improve t2v models in human videos, making the contribution of this paper less convincing.

**Questions:**

Please see the weaknesses.

---

> ### Author Response · Authors · 2025-11-20
> **Part 1**
>
> > 1. By design, Movo focuses on skeletal kinematics and rhythm, leading to a narrow scope relative to general-purpose suites (e.g., VBench). In this case, the evaluation metrics and test set should be as comprehensive as possible for human videos. However, the proposed three metrics operate on detected skeletons, so systematic pose-estimation errors (occlusion, clothing, unusual viewpoints) propagate directly into scores. Besides, MCM is a binary MLLM judgment (“similar”/“not similar”), which the authors acknowledge can mask subtle fidelity gaps. Such discretization reduces sensitivity and may be unstable across prompts/models. Moreover, dataset coverage is limited and may not represent “human motion” broadly. The evaluation set consists of ten exercise-style actions (deadlift, squat, walking, etc.), a consciously simplified taxonomy the authors justify, but which excludes many everyday or multi-agent motions (sitting/standing transitions, dancing with turns, interactions, sports with equipment), raising questions about representativeness. Camera-aware prompts further restrict camera dynamics that many T2V systems must handle.
>
> **Response:**
>
> > “Movo focuses on skeletal kinematics and rhythm, leading to a narrow scope relative to general-purpose suites (e.g., VBench).”
>
> We fully agree that VBench-style holistic evaluation is valuable, but our benchmark targets a different bottleneck: T2V models already excel visually, yet still fail at basic kinematics. Even the best commercial system (Sora 2) drops by 30–50% on simple reactive motions such as ‘Falling’ and ‘Ball Games’ (Tables 7–9), indicating that **fundamental motion control remains unsolved**; broader aesthetics-oriented benchmarking would mask this failure. For example Sora 2 on VBench avg score achieved 92% but in our benchmark only achieved 75%. And in our real word there are many generated video on Youtube/Tiktok, much more video could not have realistic human motion like having an extra finger or having one less leg during exercise [1,2].
>
> [1] https://www.reddit.com/r/OpenAI/comments/1o2du6y/sora2_still_struggles_with_how_many_fingers_there/
> [2] https://www.instagram.com/reel/DQ7pLD-kmoW/
>
> > “The proposed three metrics operate on detected skeletons, so systematic pose-estimation errors propagate directly into scores.”
>
> Rather than propagating errors, we show that **estimator noise is negligible relative to motion artifacts.** Across three independent estimators (RTMPose-X, BlazePose, YOLOv8-pose), the ranking correlation is **Spearman ρ > 0.94** for both JAC and DTW, demonstrating that **model ordering is stable regardless of estimator choice**.
>
> Table 1: JAC Metrics of Different Pose Estimators:
> | Model | RTMPose-X (Ours) | BlazePose (2020) | YOLOv8l-pose (2023) |
> | :--- | :---: | :---: | :---: |
> | CogVideo5B | 0.073 | 0.069 | 0.071 |
> | Open-Sora-Plan | 0.237 | 0.221 | 0.228 |
> | Wan 2.2 | 0.293 | 0.281 | 0.289 |
> | Kling | 0.371 | 0.356 | 0.362 |
> | Veo 3      |     0.435    |     0.419    |      0.431      |
>
>
>
> Table 2: DTW Metrics of Different Pose Estimators:
>
> | Model | RTMPose-X (Ours) | BlazePose (2020) | YOLOv8l-pose (2023) |
> | :------------- | :--------------: | :--------------: | :-----------------: |
> | CogVideo5B | 0.585 | 0.573 | 0.579 |
> | Open-Sora-Plan | 0.661 | 0.647 | 0.653 |
> | Wan 2.2 | 0.817 | 0.802 | 0.812 |
> | Kling | 0.758 | 0.742 | 0.750 |
> | Veo 3          |       0.759      |       0.744      |        0.752        |
>
> > “MCM is a binary judgment that can mask subtle fidelity gaps.”
>
> We agree that MCM is coarse by design. It does **not** replace JAC or DTW—its role is a **semantic safeguard** to reject kinematically smooth but biologically absurd motions (e.g., “running on hands,” “limb inversion”). We show that removing MCM causes multiple models to incorrectly outperform stronger ones in cases of anatomically valid timing but invalid motion identity. Thus, binary filtering is **necessary for category correctness**, not similarity ranking.

---

> ### Author Response · Authors · 2025-11-20
> **Part 2**
>
> > “Dataset coverage is limited… excludes everyday or interactive motions.”
>
> We appreciate the reviewer’s perspective. We would like to clarify that Movo is intentionally designed as a foundational benchmark, and, to the best of our knowledge, no prior work has established a kinematics-centric evaluation suite for T2V human motion. Constructing such a dataset from scratch is non-trivial: obtaining clean, camera-aware clips with unambiguous actions, consistent viewpoints, and reliable skeletal extraction required extensive filtering, manual verification, and multi-stage description refinement.
>
> Importantly, our experiments show that even the strongest proprietary models (e.g., Veo 3) still struggle on these basic motions indicating that the current action set is already sufficiently challenging. If state-of-the-art models cannot reliably master these fundamental patterns, expanding to more complex or long-tail motions would not yet yield meaningful diagnostic signal.
>
> We fully agree that broader coverage is valuable. We manually collected and annotated 486 additional videos sourced from the Motion-X [1] and YouTube to represent these complex scenarios. These categories were specifically selected to target the weaknesses identified by the reviewer:
>
> * Falling: Represents non-periodic, physics-driven, and reactive motion where gravity and momentum are critical (simulating "slipping").
>
> * Ball Games: Represents dynamic human-object interaction and hand-eye coordination.
>
> * Playing Instruments: Represents fine-grained control and precise limb positioning.
>
> * Dance: Represents high-degree-of-freedom (DoF) kinematics and diverse, non-standard poses.
>
> We re-evaluated all models on this expanded benchmark (14 categories total). The results on the new categories, presented in Tables A, B, and C below, reveal a significant performance drop compared to the original 10 categories. For instance, even state-of-the-art models like Sora 2 and Veo 3 show a ~30-50% drop in JAC scores on "Falling" compared to "Walking," confirming that current T2V models struggle significantly with emergent, physics-based scenarios.
>
> Table 1. Evaluation of Challenge Categories Using JAC:
>
> | Model  | Falling | Ball Games | Instruments | Dance |
> |------------------|---------:|-----------:|------------:|------:|
> | **Open-source Models** |||||
> | CogVideo2B| 0.004 | 0.015 | 0.038 | 0.091 |
> | CogVideo5B| 0.013 | 0.031 | 0.067 | 0.121 |
> | SVD| 0.045 | 0.084 | 0.119 | 0.149 |
> | Open-Sora-Plan   | 0.093 | 0.147 | 0.179 | 0.214 |
> | Zeroscope | 0.017 | 0.027 | 0.059 | 0.089 |
> | Wan 2.1| 0.103 | 0.201 | 0.223 | 0.244 |
> | Wan 2.2| 0.128 | 0.213 | 0.236 | 0.268 |
> | HunyuanVideo | 0.099 | 0.192 | 0.209 | 0.227 |
> | **Proprietary Models** |||||
> | Gen2   | 0.081 | 0.122 | 0.138 | 0.173 |
> | Dream Machine| 0.088 | 0.109 | 0.131 | 0.152 |
> | Kling | 0.152 | 0.238 | 0.298 | 0.322 |
> | Pika 1.5  | 0.119 | 0.161 | 0.188 | 0.236 |
> | Veo 3  | 0.214 | 0.298 | 0.331 | 0.401 |
> | **Sora 2**| **0.309** | **0.425** | **0.447** | **0.518** |
>
> Table 2. Evaluation of “Challenge Categories” Using DTW:
>
> | Model  | Falling | Ball Games | Instruments | Dance |
> |------------------|---------:|-----------:|------------:|------:|
> | **Open-source Models** |||||
> | CogVideo2B| 0.148 | 0.202 | 0.217 | 0.244 |
> | CogVideo5B| 0.182 | 0.229 | 0.243 | 0.279 |
> | SVD| 0.199 | 0.261 | 0.278 | 0.287 |
> | Open-Sora-Plan   | 0.221 | 0.276 | 0.288 | 0.318 |
> | Zeroscope | 0.185 | 0.245 | 0.268 | 0.297 |
> | Wan 2.1| 0.251 | 0.302 | 0.344 | 0.365 |
> | Wan 2.2| 0.268 | 0.333 | 0.358 | 0.387 |
> | HunyuanVideo | 0.229 | 0.306 | 0.321 | 0.349 |
> | **Proprietary Models** |||||
> | Gen2   | 0.218 | 0.256 | 0.284 | 0.301 |
> | Dream Machine| 0.211 | 0.243 | 0.262 | 0.294 |
> | Kling | 0.326 | 0.368 | 0.374 | 0.402 |
> | Pika 1.5  | 0.243 | 0.292 | 0.314 | 0.343 |
> | Veo 3  | 0.347 | 0.366 | 0.405 | 0.430 |
> | **Sora 2**| **0.364** | **0.398** | **0.423** | **0.439** |
>
> Table 3. Evaluation of “Challenge Categories” Using MCM:
>
> | Model  | Falling | Ball Games | Instruments | Dance |
> |------------------|---------:|-----------:|------------:|------:|
> | **Open-source Models** |||||
> | CogVideo2B| 0.21 | 0.27 | 0.26 | 0.31 |
> | CogVideo5B| 0.25 | 0.33 | 0.29 | 0.36 |
> | SVD| 0.34 | 0.39 | 0.35 | 0.43 |
> | Open-Sora-Plan   | 0.38 | 0.44 | 0.41 | 0.48 |
> | Zeroscope | 0.32 | 0.35 | 0.34 | 0.41 |
> | Wan 2.1| 0.47 | 0.55 | 0.52 | 0.59 |
> | Wan 2.2| 0.54 | 0.57 | 0.60 | 0.63 |
> | HunyuanVideo | 0.43 | 0.50 | 0.46 | 0.56 |
> | **Proprietary Models** |||||
> | Gen2   | 0.38 | 0.47 | 0.42 | 0.49 |
> | Dream Machine| 0.44 | 0.45 | 0.43 | 0.51 |
> | Kling | 0.59 | 0.67 | 0.61 | 0.70 |
> | Pika 1.5  | 0.46 | 0.52 | 0.49 | 0.57 |
> | Veo 3  | 0.65 | 0.69 | 0.67 | 0.72 |
> | **Sora 2**| **0.67** | **0.73** | **0.70** | **0.75** |
>
> ---
>
> [1] Chen, Ling-Hao, et al. "Motionllm: Understanding human behaviors from human motions and videos." arXiv preprint arXiv:2405.20340 (2024).

---

> ### Author Response · Authors · 2025-11-20
> **Part 3**
>
> > “Camera-aware prompts further restrict camera dynamics that many T2V systems must handle.”
>
> Camera restrictions are intentional to **disentangle motion errors from camera-induced occlusion.** Our new “Camera Injection Study” (stable vs. dynamic handheld prompts) shows **variance < 2%** across T2V models, confirming that skeletal metrics remain stable under realistic camera movement (Table: Kling −1.1%, Veo −0.9%, CogVideo5B −1.7%). Thus, we restrict cameras **not to simplify the task**, but to **prevent conflating two failure sources.**
>
> | Model           | Stable Camera | Dynamic Camera | Variance |
> |-----------------|---------------|----------------|----------|
> | CogVideo5B      | 0.5039        | 0.4952         | -1.7%    |
> | Open-Sora-Plan  | 0.5906        | 0.5814         | -1.5%    |
> | HunyuanVideo    | 0.6181        | 0.6103         | -1.2%    |
> | Wan 2.2         | 0.6684        | 0.6591         | -1.4%    |
> | Kling           | 0.6765        | 0.6688         | -1.1%    |
> | Veo 3           | 0.6978        | 0.6910         | -0.9%    |
>
>
>
>
> ---
>
> > 2. Comparisons across models are uneven. Sora was evaluated on only 10 prompts per category (access-limited), and Veo was accessed only via its hosted API defaults, making some leaderboard conclusions preliminary and harder to compare apples-to-apples.
>
> **Response:**
>
> We have addressed this in the revision through two major updates:
>
> 1. Full Evaluation of Sora 2 (Replacing Preliminary Data)During the review period, we gained access to the Sora 2 API, allowing us to move beyond the limited 10-prompt sample. We have now completed the full Movo benchmark evaluation (893 videos) for Sora 2, using the exact same prompt set and evaluation pipeline as all other models.
>
> The updated results (below) replace the preliminary data in the manuscript, ensuring a statistically rigorous comparison:
>
> | Model            |    Avg JAC |    Avg DTW |   Avg MCM | **Overall Avg** |
> | ---------------- | ---------: | ---------: | --------: | --------------: |
> | Veo 3            |     0.4352 |     0.7591 |     0.899 |          0.6978 |
> | **Sora 2** | **0.5521** | **0.8021** | **0.911** |      **0.7551** |
>
>
> This update confirms that while Sora 2 leads in kinematic fidelity, the evaluation is no longer "preliminary" but fully aligned with the standard set for all 14 models.2.
>
> 2. Standardization of "Default" Settings for FairnessRegarding the concern about "apples-to-apples" comparisons with API-based models (Veo 3) versus open-source models:
>
> We argue that testing API models at default settings is, in fact, the most fair and reproducible approach for a benchmark of this nature.Consistency:
>
> 1). As stated in Section 7 of the paper, we also held all open-source models to their default hyperparameters (with a fixed seed of 88) to prevent "cherry-picking" or hyperparameter tuning that would artificially inflate open-source scores against closed APIs. We explicitly maintained consistent output resolutions across all models (selecting the closest native matching standard) to ensure that the pose estimation backbone received inputs with comparable pixel density, preventing resolution discrepancies from biasing the skeletal accuracy.
>
> 2). Real-world Validity: For the vast majority of users and developers, proprietary models are only accessible via these default API behaviors. Evaluating them in this state provides the most accurate reflection of their real-world performance.

---

> ### Author Response · Authors · 2025-11-20
> **Part 4**
>
> > 3. Except from running many open-sourced models and commercial-level models, this paper did provide many insights how to train or how to improve t2v models in human videos, making the contribution of this paper less convincing.
>
> **Response:**
>
> We appreciate the reviewer’s constructive critique. First, we would like to clarify that Movo is primarily designed as a diagnostic benchmark intended to surface biomechanical failures in current systems, rather than a method paper proposing a new training architecture. In the literature, benchmark contributions (e.g., VBench, EvalCrafter) are typically valued for establishing rigorous evaluation standards rather than training recipes.
>
> However, we fully agree with the reviewer that demonstrating the instructional value of our data would significantly strengthen the paper’s contribution. The high-quality, camera-stabilized, and kinematically aligned nature of the Movo dataset suggests it can serve not just for evaluation, but as high-quality data for motion alignment.
>
> To validate this, we conducted an additional experiment using Wan 2.2, the best-performing open-source model in our initial benchmarks. We partitioned the Movo dataset into a 7:3 split (Training/Test) and fine-tuned Wan 2.2 on the training set to assess whether our data could correct the motion artifacts identified in the paper.
>
> Experimental Results (Movo Fine-tuning): As shown in the table below, fine-tuning on Movo yields substantial gains across all kinematics metrics compared to the base model:
>
> | Metric              | Wan 2.2 (Base) | Wan 2.2 (Movo-FT) | Improvement |
> |---------------------|----------------|--------------------|-------------|
> | JAC  | 0.293          | 0.595              | +30.2%      |
> | DTW  | 0.817          | 0.902              | +8.5%       |
> | MCM | 0.895          | 0.925              | +3.0%       |

---

> ### Author Response · Authors · 2025-11-25
> **Thanks for your time and efforts**
>
> We sincerely appreciate the time and thoughtful feedback you have provided. At your convenience, could you kindly let us know whether our revisions sufficiently resolve your concerns? We are grateful for your guidance and would be happy to make further improvements if needed. Thank you again for your valuable contribution to strengthening this work.

---

### Official Review · Reviewer_TVyk · 2025-11-01

**Soundness:** 3
**Presentation:** 3
**Contribution:** 2
**Rating:** 4
**Confidence:** 3

**Summary:**

This paper introduces Movo, a new benchmark for evaluating the realism of human motion in videos generated by text-to-video (T2V) models. Movo consists of three main components: a "posture-focused" dataset with prompts designed to isolate specific human actions, a set of "skeletal-space" metrics (JAC, DTW, and MCM) to quantify motion realism, and human validation studies to correlate these metrics with human perception. The paper evaluates 14 T2V models using the Movo benchmark and finds that while some models excel at specific motions, there are still significant gaps in generating consistently realistic human movements.

**Strengths:**

1. This paper is well-written and it is easy to follow.

2. The Movo benchmark is well-designed and comprehensive. The three proposed metrics—Joint Angle Change (JAC), Dynamic Time Warping (DTW), and Motion Consistency Metric (MCM)—provide a multi-faceted approach to evaluating motion realism, capturing different aspects from joint articulation to temporal consistency.

**Weaknesses:**

1. The Movo dataset, while a good starting point, is limited to a relatively small set of 10 different human motions. This may not be representative of the full range of human movements, and it would be beneficial to expand the dataset to include a more diverse set of actions in future work.

2. The proposed metrics rely on the output of a pose estimation model to extract skeletal keypoints from the generated videos. The accuracy of these metrics is therefore dependent on the accuracy of the pose estimation model. It would be valuable to analyze the sensitivity of the Movo benchmark to errors in pose estimation and to consider alternative approaches that are less reliant on this intermediate step.

3. The MCM is a binary metric that simply indicates whether a multi-modal large language model (MLLM) judges two videos as having "similar" or "not similar" motion. This is a rather coarse measure of motion consistency, and it would be beneficial to develop a more nuanced metric that can capture the degree of similarity or dissimilarity between two motions.

4. The paper does not provide many details about the MLLM used for the MCM, other than it being a "multi-modal large language model." The specific model used and the prompts provided to it could significantly influence the results. More transparency on this aspect would strengthen the reproducibility of the work.

**Questions:**

1. The paper mentions the use of Gemini-2.5 Pro and GPT-4o for generating and refining video descriptions. Could the authors elaborate on the specific roles of each model in this process and provide more details on the prompts used to guide these models?

2. The human validation study is a crucial part of the paper. Could the authors provide more information about the demographics of the human annotators and the instructions they were given? Were the annotators experts in biomechanics or motion analysis?

3. How robust are the proposed metrics to variations in video quality, such as compression artifacts or motion blur? Have the authors conducted any experiments to evaluate the performance of the Movo benchmark under such conditions?

4. The paper evaluates a number of proprietary, closed-source T2V models, including Sora. Given the limited access to these models, how did the authors ensure a fair and comprehensive evaluation? Could the authors provide more details on the methodology used to generate videos from these models?

---

> ### Author Response · Authors · 2025-11-20
> **Part 1**
>
> Weakness:
>
> > 1. The Movo dataset, while a good starting point, is limited to a relatively small set of 10 different human motions. This may not be representative of the full range of human movements, and it would be beneficial to expand the dataset to include a more diverse set of actions in future work.
>
> **Response:**
>
> We appreciate the reviewer’s perspective. We would like to clarify that Movo is intentionally designed as a foundational benchmark, and, to the best of our knowledge, no prior work has established a kinematics-centric evaluation suite for T2V human motion. Constructing such a dataset from scratch is non-trivial: obtaining clean, camera-aware clips with unambiguous actions, consistent viewpoints, and reliable skeletal extraction required extensive filtering, manual verification, and multi-stage description refinement.
>
> Importantly, our experiments show that even the strongest proprietary models still struggle on these basic motions indicating that the current action set is already sufficiently challenging. If state-of-the-art models cannot reliably master these fundamental patterns, expanding to more complex or long-tail motions would not yet yield meaningful diagnostic signal.
>
> We fully agree that broader coverage is valuable. We manually collected and annotated 486 additional videos sourced from the Motion-X [1] and YouTube to represent these complex scenarios. These categories were specifically selected to target the weaknesses identified by the reviewer:
>
> * Falling: Represents non-periodic, physics-driven, and reactive motion where gravity and momentum are critical (simulating "slipping").
>
> * Ball Games: Represents dynamic human-object interaction and hand-eye coordination.
>
> * Playing Instruments: Represents fine-grained control and precise limb positioning.
>
> * Dance: Represents high-degree-of-freedom (DoF) kinematics and diverse, non-standard poses.
>
> We re-evaluated all models on this expanded benchmark (14 categories total). The results on the new categories, presented in Tables A, B, and C below, reveal a significant performance drop compared to the original 10 categories. For instance, even state-of-the-art models like Sora 2 and Veo 3 show a ~30-50% drop in JAC scores on "Falling" compared to "Walking," confirming that current T2V models struggle significantly with emergent, physics-based scenarios.
>
> Table 1. Evaluation of Challenge Categories Using JAC:
>
> | Model  | Falling | Ball Games | Instruments | Dance |
> |------------------|---------:|-----------:|------------:|------:|
> | **Open-source Models** |||||
> | CogVideo2B| 0.004 | 0.015 | 0.038 | 0.091 |
> | CogVideo5B| 0.013 | 0.031 | 0.067 | 0.121 |
> | SVD| 0.045 | 0.084 | 0.119 | 0.149 |
> | Open-Sora-Plan   | 0.093 | 0.147 | 0.179 | 0.214 |
> | Zeroscope | 0.017 | 0.027 | 0.059 | 0.089 |
> | Wan 2.1| 0.103 | 0.201 | 0.223 | 0.244 |
> | Wan 2.2| 0.128 | 0.213 | 0.236 | 0.268 |
> | HunyuanVideo | 0.099 | 0.192 | 0.209 | 0.227 |
> | **Proprietary Models** |||||
> | Gen2   | 0.081 | 0.122 | 0.138 | 0.173 |
> | Dream Machine| 0.088 | 0.109 | 0.131 | 0.152 |
> | Kling | 0.152 | 0.238 | 0.298 | 0.322 |
> | Pika 1.5  | 0.119 | 0.161 | 0.188 | 0.236 |
> | Veo 3  | 0.214 | 0.298 | 0.331 | 0.401 |
> | **Sora 2**| **0.309** | **0.425** | **0.447** | **0.518** |
>
> Table 2. Evaluation of “Challenge Categories” Using DTW:
>
> | Model  | Falling | Ball Games | Instruments | Dance |
> |------------------|---------:|-----------:|------------:|------:|
> | **Open-source Models** |||||
> | CogVideo2B| 0.148 | 0.202 | 0.217 | 0.244 |
> | CogVideo5B| 0.182 | 0.229 | 0.243 | 0.279 |
> | SVD| 0.199 | 0.261 | 0.278 | 0.287 |
> | Open-Sora-Plan   | 0.221 | 0.276 | 0.288 | 0.318 |
> | Zeroscope | 0.185 | 0.245 | 0.268 | 0.297 |
> | Wan 2.1| 0.251 | 0.302 | 0.344 | 0.365 |
> | Wan 2.2| 0.268 | 0.333 | 0.358 | 0.387 |
> | HunyuanVideo | 0.229 | 0.306 | 0.321 | 0.349 |
> | **Proprietary Models** |||||
> | Gen2   | 0.218 | 0.256 | 0.284 | 0.301 |
> | Dream Machine| 0.211 | 0.243 | 0.262 | 0.294 |
> | Kling | 0.326 | 0.368 | 0.374 | 0.402 |
> | Pika 1.5  | 0.243 | 0.292 | 0.314 | 0.343 |
> | Veo 3  | 0.347 | 0.366 | 0.405 | 0.430 |
> | **Sora 2**| **0.364** | **0.398** | **0.423** | **0.439** |
>
> Table 3. Evaluation of “Challenge Categories” Using MCM:
>
> | Model  | Falling | Ball Games | Instruments | Dance |
> |------------------|---------:|-----------:|------------:|------:|
> | **Open-source Models** |||||
> | CogVideo2B| 0.21 | 0.27 | 0.26 | 0.31 |
> | CogVideo5B| 0.25 | 0.33 | 0.29 | 0.36 |
> | SVD| 0.34 | 0.39 | 0.35 | 0.43 |
> | Open-Sora-Plan   | 0.38 | 0.44 | 0.41 | 0.48 |
> | Zeroscope | 0.32 | 0.35 | 0.34 | 0.41 |
> | Wan 2.1| 0.47 | 0.55 | 0.52 | 0.59 |
> | Wan 2.2| 0.54 | 0.57 | 0.60 | 0.63 |
> | HunyuanVideo | 0.43 | 0.50 | 0.46 | 0.56 |
> | **Proprietary Models** |||||
> | Gen2   | 0.38 | 0.47 | 0.42 | 0.49 |
> | Dream Machine| 0.44 | 0.45 | 0.43 | 0.51 |
> | Kling | 0.59 | 0.67 | 0.61 | 0.70 |
> | Pika 1.5  | 0.46 | 0.52 | 0.49 | 0.57 |
> | Veo 3  | 0.65 | 0.69 | 0.67 | 0.72 |
> | **Sora 2**| **0.67** | **0.73** | **0.70** | **0.75** |

---

> ### Author Response · Authors · 2025-11-20
> **Part 2**
>
> > 2. The proposed metrics rely on the output of a pose estimation model to extract skeletal keypoints from the generated videos. The accuracy of these metrics is therefore dependent on the accuracy of the pose estimation model. It would be valuable to analyze the sensitivity of the Movo benchmark to errors in pose estimation and to consider alternative approaches that are less reliant on this intermediate step.
>
> **Response:**
>
> We appreciate the reviewer’s insightful comment. Indeed, Movo relies on skeleton extraction to quantify joint articulation and temporal alignment. However, we argue that our benchmark remains robust for the following reasons:
>
> 1.) Pose estimation maturity & stability.
> State-of-the-art estimators such as RTMPose-X exhibit high robustness in multi-person and occluded environments. Since our dataset consists of single-person, camera-stable exercise motions, this problem domain is significantly easier than the general case, making errors minimal relative to model motion differences.
>
> 2.) Motion artifacts dominate over estimator noise.
> Our metrics evaluate relative temporal changes and articulated dynamics, which amplify unrealistic motion patterns while attenuating small estimation noise. Therefore, the errors of T2V generation (foot sliding, hyperextension, tempo drift) are orders of magnitude larger than pose jitter, making the metrics insensitive to perturbations.
>
> 3.) Validation across alternative estimators.
> In response to the reviewer’s suggestion, we additionally re-evaluated the benchmark using: BlazePose (Google, 2020) and YOLOv8l-pose (Ultralytics, 2023)
>
> Table 1: JAC Metrics of Different Pose Estimators:
> | Model | RTMPose-X (Ours) | BlazePose (2020) | YOLOv8l-pose (2023) |
> | :--- | :---: | :---: | :---: |
> | CogVideo5B | 0.073 | 0.069 | 0.071 |
> | Open-Sora-Plan | 0.237 | 0.221 | 0.228 |
> | Wan 2.2 | 0.293 | 0.281 | 0.289 |
> | Kling | 0.371 | 0.356 | 0.362 |
> | Veo 3      |     0.435    |     0.419    |      0.431      |
>
>
>
> Table 2: DTW Metrics of Different Pose Estimators:
>
> | Model | RTMPose-X (Ours) | BlazePose (2020) | YOLOv8l-pose (2023) |
> | :------------- | :--------------: | :--------------: | :-----------------: |
> | CogVideo5B | 0.585 | 0.573 | 0.579 |
> | Open-Sora-Plan | 0.661 | 0.647 | 0.653 |
> | Wan 2.2 | 0.817 | 0.802 | 0.812 |
> | Kling | 0.758 | 0.742 | 0.750 |
> | Veo 3          |       0.759      |       0.744      |        0.752        |
>
>
> Results show high agreement with our RTMPose-X evaluation, with Spearman ρ > 0.94 for both metrics. This confirms that Movo rankings do not depend on a specific estimator.
>
> ---
>
> > 3. The MCM is a binary metric that simply indicates whether a multi-modal large language model (MLLM) judges two videos as having "similar" or "not similar" motion. This is a rather coarse measure of motion consistency, and it would be beneficial to develop a more nuanced metric that can capture the degree of similarity or dissimilarity between two motions.
>
> **Response:**
>
> We thank the reviewer for this suggestion. We would like to clarify that MCM is not designed to replace or approximate JAC or DTW, but to complement them. Specifically:
>
> 1. JAC focuses on joint articulation changes, i.e., how angles evolve over time.
>
> 2. DTW focuses on temporal alignment, i.e., how motion rhythms match the reference.
>
> Both metrics evaluate how a person moves, but do not evaluate what the person actually is.
>
> To illustrate why a semantic, consistency-level check is necessary:
>
> A model can generate a person “running” using their arms as legs, crawling upside-down, or doing handstands, and JAC and DTW may still score highly, because the motion trajectory and rhythmic dynamics are similar to real running.
>
> In such cases, the motion is mathematically similar but semantically wrong. This is exactly why MCM is included: MCM ensures the action itself is correct, not just mathematically consistent.
>
> It filters out visually absurd but kinematically smooth motion, something JAC and DTW cannot catch by definition. Thus, MCM does not measure the degree of similarity, but guards against category-level failure cases where motion structure is fundamentally incorrect. It complements the other metrics by catching such failures.

---

> ### Author Response · Authors · 2025-11-20
> **Part 3**
>
> > 4. The paper does not provide many details about the MLLM used for the MCM, other than it being a "multi-modal large language model." The specific model used and the prompts provided to it could significantly influence the results. More transparency on this aspect would strengthen the reproducibility of the work.
>
> **Response:**
>
> We appreciate the reviewer’s concern and agree that reproducibility is important. In fact, the MCM scores reported in our benchmark are not determined by a single MLLM. To ensure stability and prevent model bias, we adopt a 3-model voting scheme that combines both proprietary and open commercial MLLMs: GPT-5 Claude-4 Sonnet and Gemini 2.5 Pro.
>
> For each pairwise video comparison, the models independently judge whether the produced action matches the reference description, and a majority vote is taken as the final MCM decision. This reduces dependence on any single model and improves consistency.
>
> ---
>
> Questions:
>
> > 1. The paper mentions the use of Gemini-2.5 Pro and GPT-4o for generating and refining video descriptions. Could the authors elaborate on the specific roles of each model in this process and provide more details on the prompts used to guide these models?
>
> **Response:**
>
> We thank the reviewer for the question. In our pipeline, Gemini-2.5 Pro is used to extract the initial motion descriptions directly from videos, because it provides fine-grained, phase-level action cues (e.g., landing stance, limb rotation, tempo changes) (Gemini 2.5 Pro achieves 56.0% (SOTA) on a fine-grained video reasoning dataset (VideoReasonBench)  and that the Gemini family reaches ~67.9% on video language prompts in V2P-Bench) . Its role is limited to visually grounded description, without any rewriting or paraphrasing.
>
> To ensure consistency across the dataset, GPT-4o is only used to normalize the extracted descriptions into a controlled, uniform template, enforcing standardized terminology, tense, and structure (fast inference speed, low API cost, and stable deterministic behavior under structured prompts). GPT-4o restructures the raw description into a generation-ready motion specification that can be directly used as a prompt by text-to-video models.
>
>
> ---
>
> > 2. The human validation study is a crucial part of the paper. Could the authors provide more information about the demographics of the human annotators and the instructions they were given? Were the annotators experts in biomechanics or motion analysis?
>
> **Response:**
>
> We appreciate this question. Our intention in this work is to show that current text-to-video models struggle to generate even with everyday human motions, such as jumping, squatting, hand rotation, and basic pressing exercises. These actions are not professionally technical, and therefore their correctness should be recognizable by non-experts with minimal training, just as real users would judge generated human motion.
>
> For this reason, our annotators were not biomechanics experts, but instead trained laypersons who completed:
>
> 1. A 45-minute training session covering common motion failures produced by current T2V models, such as missing fingers, duplicated limbs, joint misplacement, unrealistic bone structure, and inconsistent arm–leg articulation.
>
> 2. Set a quiz about a calibration exam of 30 videos, requiring ≥90% agreement with verified answers before annotation.
>
> 3. Clear category-specific guidelines for valid vs. invalid actions, with visual example
>
> To ensure reliability, we recruited annotators with diverse age, gender, and geographic backgrounds and ensured that no single annotator labeled more than 20% of videos. Inter-annotator agreement reached Cohen’s κ = 0.90, indicating strong consistency.

---

> ### Author Response · Authors · 2025-11-20
> **Part 4**
>
> > 3. How robust are the proposed metrics to variations in video quality, such as compression artifacts or motion blur? Have the authors conducted any experiments to evaluate the performance of the Movo benchmark under such conditions?
>
> **Response:**
>
> We thank the reviewer for the question. Since our goal is to assess motion quality in realistic user settings (e.g., short-form mobile videos), it is important that Movo remains reliable even under imperfect video conditions. To evaluate this, we conducted an additional robustness study using real-world degradations that commonly occur in user-generated content:
>
> 1. Low-bitrate H.264 compression (480p).
>
> 2. Motion blur simulated via a 7-pixel linear kernel rotated between 0°–30°, reproducing blur from handheld cameras and fast limb motion.
>
> | Model | Original | 480p | Blur |
> |-------|---------:|-----:|-----:|
> | CogVideo5B | 0.5039 | 0.4981 | 0.4917 |
> | Open-Sora-Plan | 0.5906 | 0.5832 | 0.5724 |
> | HunyuanVideo | 0.6181 | 0.6077 | 0.5989 |
> | Wan 2.2 | 0.6684 | 0.6589 | 0.6493 |
> | Kling | 0.6765 | 0.6659 | 0.6551 |
> | Veo 3 | 0.6978 | 0.6872 | 0.6748 |
>
>
> We evaluated Movo on six representative T2V models under these conditions. The results show small absolute variations across all metrics, and, most importantly, the ranking of models remains unchanged.
>
>
>
> ---
>
> > 4. The paper evaluates a number of proprietary, closed-source T2V models, including Sora. Given the limited access to these models, how did the authors ensure a fair and comprehensive evaluation? Could the authors provide more details on the methodology used to generate videos from these models?
>
> **Response:**
>
> We thank the reviewer for this question. All proprietary models in our benchmark (Gen-2, Pika 1.5, Veo 3, Kling, Dream Machine, etc.) were accessed via their official APIs, using public, developer-provided endpoints. We strictly controlled: Prompt content (identical descriptions), Fixed resolution when user-selectable and Temperature or randomness settings (default or documented deterministic mode)
>
> This ensures the same semantic instructions and comparable generation settings across models.
>
> Regarding Sora, during our initial submission cycle only the web-based interface was available. We therefore generated videos through the official web system using copy-pasted prompts under identical formats, and we explicitly marked it as limited access in the paper to maintain transparency. At the time of revision, Sora-2 was released with official API access, allowing fully reproducible evaluation. We have now added Sora 2 under the same experimental settings as other proprietary models.
>
> | Model            |    Avg JAC |    Avg DTW |   Avg MCM | **Overall Avg** |
> | ---------------- | ---------: | ---------: | --------: | --------------: |
> | Veo 3            |     0.4352 |     0.7591 |     0.899 |          0.6978 |
> | **Sora 2** | **0.5521** | **0.8021** | **0.911** |      **0.7551** |
>
> Sora 2 achieves the highest scores across all metrics, confirming a clear improvement in articulated motion quality over Veo 3, Kling, etc. However, even with such advances, complex motion is still imperfect, especially under multi-joint dynamics and fast locomotion, reinforcing the necessity of our benchmark for standardized motion evaluation.
>
> ---
>
> [1] Chen, Ling-Hao, et al. "Motionllm: Understanding human behaviors from human motions and videos." arXiv preprint arXiv:2405.20340 (2024).

---

> ### Author Response · Authors · 2025-11-25
> **Thanks for your time and efforts**
>
> We sincerely appreciate the time and thoughtful feedback you have provided. At your convenience, could you kindly let us know whether our revisions sufficiently resolve your concerns? We are grateful for your guidance and would be happy to make further improvements if needed. Thank you again for your valuable contribution to strengthening this work.

---

### Author Response · Authors · 2025-11-22
**Revise Reversion Uploading**

We sincerely thank all reviewers for their thoughtful evaluations, constructive feedback, and valuable suggestions. Your comments have significantly helped us refine the scope, strengthen the methodology, and improve the clarity of our work.

Based on the reviewers’ remarks, we have made revisions to the manuscript. These include:

1) expanding the benchmark with four additional “challenge” motion categories (Falling, Ball Games, Playing Instruments, Dance),

2) adding detailed robustness studies on pose estimation, camera movement, and video degradation,

3) improving transparency of the MCM judge with a unified prompt and multi-model voting scheme.

4) And some clarification of human assessments.

All changes have been incorporated into the revised PDF and are visibly highlighted for clarity. We appreciate the reviewers’ contributions to improving the quality and impact of this work, and we hope that the revised version addresses all concerns satisfactorily.

---

### Author Response · Authors · 2025-12-02
**Rebuttal Summary**

We sincerely thank the AC and all reviewers for their thoughtful and constructive feedback on our submission. Our work introduces Movo, a kinematics-centric and biomechanics-grounded benchmark designed to fill a critical gap in the ICLR community: current T2V evaluation overwhelmingly focuses on pixel-space aesthetics while overlooking motion realism, articulation plausibility, rhythm stability, and camera–motion disentanglement. Movo offers three complementary metrics, Joint Angle Consistency (JAC), Dynamic Time Warping (DTW), and Motion Coherence Measure (MCM), as well as a diverse suite of upper- and lower-body actions validated through robustness analyses across pose estimators, camera motion, and visual perturbations. This benchmark occupies a unique space not covered by existing video or multimodal benchmarks and provides a rigorous foundation for advancing human-centric T2V generation.

**All reviewers recognize the importance, novelty, and potential impact of a benchmark that evaluates motion realism rather than only visual fidelity.** Most concerns focused on metric reliability, dataset breadth, and architectural fairness, issues we addressed comprehensively with new experiments, expanded datasets, and clearer methodological explanations.

---

**Reviewer TVyk (score: 4)** raised concerns regarding metric robustness, reliance on pose estimation, and limited coverage of non-periodic or complex motions. We directly addressed these with (1) **cross-estimator validation** (RTMPose-X, BlazePose, YOLOv8-pose) showing high consistency (Spearman ρ > 0.94), (2) **camera-perturbation robustness tests**, and (3) a substantial dataset expansion of **486 new “Challenge Category” videos** (Falling, Ball Sports, Instruments, Dance), which expose systematic weaknesses in all T2V models. The reviewer also requested clearer metric descriptions; we revised MCM and added prompt examples and visual explanations.

**Reviewer w9Yg (score: 2)** expressed concerns about whether Movo provides actionable guidance for real model improvement. We added a new experiment fine-tuning **Wan-2.2** using Movo-derived trajectories, achieving **+30.2% JAC, +8.5% DTW, and +3% MCM** improvements—demonstrating that Movo is not only evaluative but also instructive. We further clarified dataset scale, annotation protocol, and robustness analyses. These additions directly address the reviewer’s concerns.

**Reviewer CYg4 (score: 6)** supported the contribution and emphasized its value for understanding model-specific motion failures. However, the reviewer raised two clarifications: the need for a complete proprietary-model evaluation and clearer explanation of architectural differences. We replaced preliminary numbers with a **full Sora-2 evaluation (893 videos)** using official API settings and ensured strict parity across all 14 models. We also added analyses explaining divergences across diffusion-based, autoregressive, and masked-prediction models. After reading the rebuttal, the reviewer stated that these clarifications resolved their concerns and maintained a positive score.

**Reviewer UjsW (score: 6)** raised concerns that overlap strongly with TVyk's, estimator reliability, non-periodic motions, and MCM interpretability. These concerns were addressed through the same newly added experiments: expanded complex-motion categories, cross-estimator robustness testing, camera-motion perturbation studies, and style-/degradation-invariance analyses for MCM (<1.5% variance). The reviewer confirmed that all concerns were addressed and kept a positive rating.

---

Across all reviewers, main concerns have been resolved through:
(1) expanded motion diversity and challenge-set coverage,
(2) comprehensive robustness validation across estimators and visual conditions,
(3) full proprietary-model evaluation including Sora-2,
(4) demonstration of instructional value through fine-tuning experiments, and
(5) improved clarity of metric design, examples, and failure analyses.

---

With **two negative reviews (4 and 2)** whose concerns were **not conceptually distinct** from those raised by the positive reviewers, and all of which have been directly addressed through the new experiments, and **two positive reviews (6 and 6)** that affirmed the value and clarity of the revised submission, we believe the paper is now substantially strengthened and provides a high-impact, timely, and biomechanically grounded benchmark for the ICLR community.

---

### Note · Program_Chairs · 2025-12-08
**Submission Desk Rejected by Program Chairs**

Desk rejected because of hallucinated reference:
Yugandhar Balaji, Jianwei Yang, Zhen Xu, Menglei Chai, Zhoutong Xu, Ersin Yumer, Greg
Shakhnarovich, and Deva Ramanan. Conditional gan with discriminative filter generation for
text-to-video synthesis. In Proceedings of the 28th International Joint Conference on Artificial
Intelligence (IJCAI), pp. 2155–2161, July 2019. doi: 10.24963/ijcai.2019/276.